# The architecture of EGFR's basal complexes reveals autoinhibition mechanisms in dimers and oligomers

Laura C. Zanetti-Domingues[1], Dimitrios Korovesis[1], Sarah R. Needham[1], Christopher J. Tynan[1], Shiori Sagawa[2], Selene K. Roberts[1], Antonija Kuzmanic[3], Elena Ortiz-Zapater[4], Purvi Jain[5], Rob C. Roovers[6], Alireza Lajevardipour [7], Paul M.P. van Bergen en Henegouwen[5], George Santis[4], Andrew H.A. Clayton[7], David T. Clarke[1], Francesco L. Gervasio [3], Yibing Shan[2], David E. Shaw[2,8], Daniel J. Rolfe[1], Peter J. Parker[9,10] & Marisa L. Martin-Fernandez[1]

Our current understanding of epidermal growth factor receptor (EGFR) autoinhibition is based on X-ray structural data of monomer and dimer receptor fragments and does not explain how mutations achieve ligand-independent phosphorylation. Using a repertoire of imaging technologies and simulations we reveal an extracellular head-to-head interaction through which ligand-free receptor polymer chains of various lengths assemble. The architecture of the head-to-head interaction prevents kinase-mediated dimerisation. The latter, afforded by mutation or intracellular treatments, splits the autoinhibited head-to-head polymers to form stalk-to-stalk flexible non-extended dimers structurally coupled across the plasma membrane to active asymmetric tyrosine kinase dimers, and extended dimers coupled to inactive symmetric kinase dimers. Contrary to the previously proposed main autoinhibitory function of the inactive symmetric kinase dimer, our data suggest that only dysregulated species bear populations of symmetric and asymmetric kinase dimers that coexist in equilibrium at the plasma membrane under the modulation of the C-terminal domain.

[1] Central Laser Facility, Research Complex at Harwell, STFC Rutherford Appleton Laboratory, Harwell Oxford, Didcot, Oxford OX11 0QX, UK. [2] D. E. Shaw Research, New York, NY 10036, USA. [3] Department of Chemistry, Faculty of Maths & Physical Sciences, University College London, London WC1H 0AJ, UK. [4] Peter Gore Department of Immunobiology, School of Immunology & Microbial Sciences, Kings College London, London SE1 9RT, UK. [5] Division of Cell Biology, Science Faculty, Department of Biology, Utrecht University, Utrecht 3584 CH, The Netherlands. [6] Merus, LSI, Yalelaan 62, 3584 CM Utrecht, The Netherlands. [7] Centre for Micro-Photonics, Faculty of Science, Engineering and Technology, Swinburne University of Technology, Hawthorn, VIC 3122, Australia. [8] Department of Biochemistry and Molecular Biophysics, Columbia University, New York, NY 10032, USA. [9] Protein Phosphorylation Laboratory, The Francis Crick Institute, 1 Midland Road, London NW 1 1AT, UK. [10] School of Cancer and Pharmaceutical Sciences, King's College London, New Hunt's House, Guy's Campus, London SE1 1UL, UK. These authors contributed equally: Laura C. Zanetti-Domingues, Dimitrios Korovesis, Sarah R. Needham. Correspondence and requests for materials should be addressed to M.L.M.-F. (email: marisa.martin-fernandez@stfc.ac.uk)

The epidermal growth factor receptor (EGFR or HER1/ErbB1) is the founding member of the human EGFR tyrosine kinase family (HER2/ErbB2/Neu, HER3/ErbB3, and HER4/ErbB4)[1]. EGFR plays a fundamental signalling role in cell growth and is frequently hyper-activated in human cancers via mutation and/or overexpression[2]. This driving role in malignancy has made EGFR a key target for anti-cancer therapy[3,4].

An EGFR monomer consists of an N-terminal ligand-binding extracellular module (ECM) connected to an intracellular module (ICM) by a single-pass transmembrane (TM) helix (Fig. 1a). The ECM comprises four domains (DI–DIV) and adopts a tethered conformation via an interaction between DII and DIV[5]. The ICM includes a short juxtamembrane (JM) segment, a tyrosine kinase domain (TKD) and a disordered carboxy-terminal region, locus of the key tyrosine phosphorylation sites[6,7]. Ligand binding stabilises the extended conformation of the ECM promoting the formation of back-to-back dimers[8,9] (Fig. 1a). Subsequent EGFR signalling across the plasma membrane depends on an allosteric interaction between an activator and receiver kinase effected

through an asymmetric TKD (aTKD) dimer[10]. Signal transduction also requires ligand-bound EGFR oligomers[11,12] formed by face-to-face interactions between back-to-back dimers[12] (Fig. 1b).

Evidence has accumulated over the years for ligand-free EGFR dimers and oligomers (see e.g. refs. [13–21]). However, the mechanisms by which ligand-independent activation of non-monomers is prevented remain unclear. Nonetheless, it is widely believed that autoinhibition is related to the adoption of an inactive symmetric TKD (sTKD) dimer revealed by X-ray structures of EGFR TKDs bearing the V924R (or V948R) and I682Q mutations at the C-lobe and N-lobe, which inhibit aTKD dimer formation (PDB ID 3GT8 (ref. [22]), 2GS7 (ref. [10]), and 5CNN (ref. [6])). The sTKD was putatively associated to a speculative side-to-side ECM tethered dimer[20] (Fig. 1c), presumably because this would provide a fail-safe approach to autoinhibition. Alternatively, molecular dynamics (MD) simulations[23] suggested that the sTKD dimer is coupled via a C-crossing TM domain dimer to a ligand-free back-to-back dimer analogous to the X-ray structure of the *Drosophila* ECM dimer[24] and a model based on

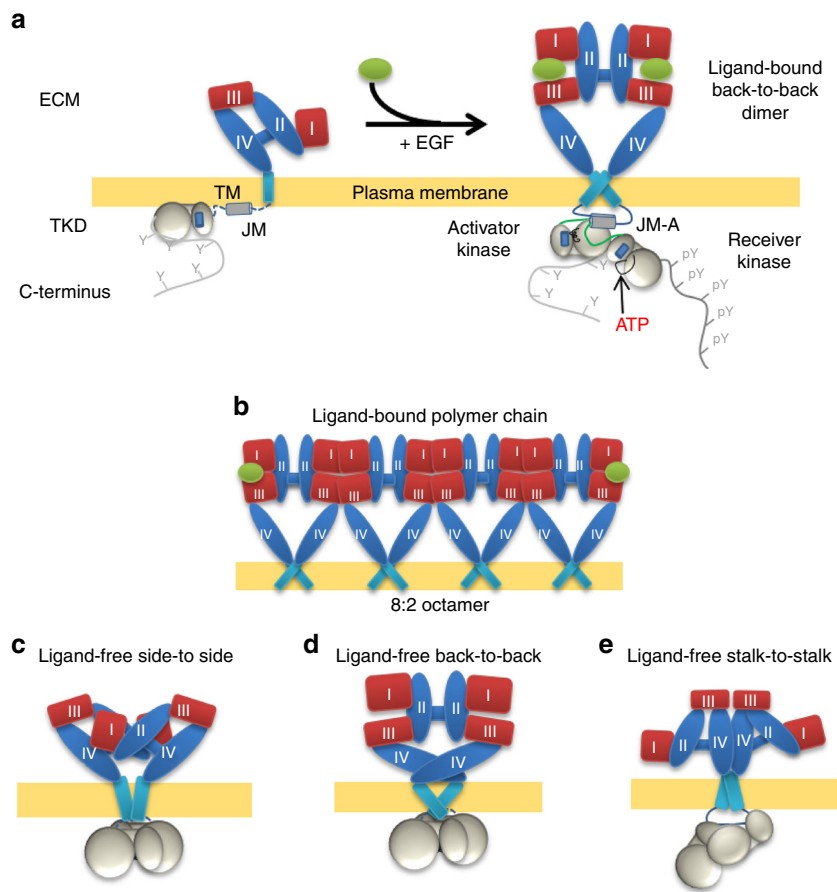

**Fig. 1** Models of ligand-free and ligand-bound EGFR complexes. **a** Top left: Cartoon of an EGFR monomer[5]. Top right: A ligand-bound back-to-back extracellular dimer[8,9]. This is linked to the catalytically active asymmetric TKD (aTKD) dimer[10] by an N-terminal crossing transmembrane (TM) dimer[40] and an antiparallel juxtamembrane-A (JM-A) helical dimer[22]. **b** Cartoon of the extracellular portion and TM domains of ligand-bound EGFR polymers formed by alternating back-to-back and face-to-face interfaces[12]. Two EGF molecules are bound at the end-receptors capping the polymer chain with a 2N:2 receptor/ligand stoichiometry. An 8:2 octamer is shown (intracellular regions not depicted). **c** Cartoon of a speculative ligand-free side-to-side dimer that would putatively combine the double autoinhibition of a tethered extracellular domain and a symmetric tyrosine kinase domain (sTKD) dimer[5,20,22]. **d** Cartoon of a ligand-free extended back-to-back dimer coupled via a TM domain C-crossing dimer to an sTKD dimer (modified from Arkhipov et al.[23]). **e** Cartoon of a stalk-to-stalk tethered dimer coupled via an N-crossing TM domain dimer to the aTKD dimer induced by TKI binding in the C-terminal domain truncated Δ998-EGFR (modified from Lu et al.[26]). For all panels ECM domains I and III are in red, II and IV in blue, EGF ligand is in green, plasma membrane in yellow, TM in teal, JM in dark grey, TKD in light grey

SAXS data from *Caenorhabditis elegans* EGFR[16] (Fig. 1d). In this back-to-back dimer, which resembles the ligand-bound dimer, the autoinhibitory heavy lifting would be done by the sTKD dimer alone[25]. A flexible ECM dimer held by DIV–DIV contacts by the plasma membrane was also suggested by electron microscopy (EM) images of purified, near-full-length Δ998-EGFR[26,27] (Fig. 1e). This stalk-to-stalk dimer is promoted by kinase-mediated interactions in response to the binding of type I tyrosine kinase inhibitors (TKIs), which reversibly bind the ATP-binding pocket stabilising the aTKD dimer[26] inhibiting C-terminal phosphorylation. The challenge is the lack of high-resolution methods on cells, which has made it impossible so far to obtain evidence on the architecture of non-monomer ligand-free species.

Here we exploit the nanoscale resolution of fluorophore localisation imaging with photobleaching (FLImP)[12,28–30] in a multi-technique study including two-colour single particle tracking (SPT)[31], fluorescence resonance energy transfer (FRET)[32,33], and MD simulations[23,34] to investigate ligand-free non-monomer species on cells. The results reveal structural insights on the inactive species and on how mutations circumvent the auto-inhibition of the basal state.

## Results

**Monomers and dimers and oligomers populate the cell surface.** We used FLImP on Chinese Hamster Ovary (CHO) cells expressing ~$10^5$ wild-type EGFR (wtEGFR) copies/cell, maintained by an inducible Tet-ON promoter[35] (Fig. 2a). FLImP measures pairwise lateral separations between identical fluorophores with ~5–7 nm resolution, returning normalised distributions in the 0–60 nm range, which is ideal to investigate EGFR dimers and oligomers[12]. We conjugated CF640R fluorophore to the single cysteine of an anti-EGFR Affibody which binds DIII of EGFR's ECM without activating the receptor[36] and accumulated a FLImP histogram of DIII–DIII separations from ligand-free wtEGFR complexes. Bayesian parameter estimation with Bayesian Information Criterion (BIC)[12] objectively decomposed the FLImP histogram into a sum of five peak components at 5, 13, 22, 30, and 46 nm (Fig. 2b). An EGFR dimer has two DIIIs and must give rise to a unique DIII–DIII separation. This must be <~20 nm, given the <3.7 nm size of Affibody probes[37] and the ~5–8 nm size of the EGFR monomer[5,8,9]. Multiple separations >20 nm and periodic DIII–DIII components therefore report ligand-free wtEGFR oligomers larger than dimers and stoichiometric extracellular interactions, respectively. These results agree with previous data from T47D cells[28], which express 10-fold fewer EGFR copies/cell, indicating that the ligand-free structures formed by EGFR are independent of receptor expression.

Being a pairwise measurement, FLImP is insensitive to monomers. To estimate the relative proportions of monomers and other species, we used model-free photobleaching image correlation spectroscopy (pbICS)[38] analysis, which we applied to CHO cells expressing wtEGFR labelled with an Alexa 488-Affibody fluorescent derivative (Fig. 2c). The results suggest that ~65% of receptors are incorporated into dimers and larger oligomers, consistent with previous estimates[18,21,39].

**A structural model of extracellular ligand-free dimers.** The nanoscale proximity between receptors in oligomers translates into a local surface density of ~5000 receptors/μm². This is above that on cells overexpressing >2 × 10⁶ EGFR copies/cell (~2000 receptors/μm²), where significant phosphorylation occurs in the absence of ligand[40]. In contrast, CHO cells expressing 20-fold fewer receptors have low basal phosphorylation[36] despite ~65% of receptors forming non-monomer complexes. This suggests that

ligand-independent phosphorylation is inhibited in ligand-free complexes, leaving two questions: what is their architecture and how is autoinhibition achieved?

Among possibilities, we considered cyclic shapes[41] and polymer chains[12] (Fig. 2d). Both architectures, when probed with a 1:1 binder like Affibody, predict a decreasing "ladder-like" DIII–DIII separation intensity consistent with FLImP results. However, the longest DIII–DIII separation (46 nm) (Fig. 2b) and largest oligomer size (octamers) (Fig. 2c) are more consistent with polymer chains. To test this further, we conjectured that, when probed with a CF640R-EGF derivative, which only binds the polymer ends[12], each polymer chain would contribute a single separation (Fig. 2d, right). Thus, the FLImP separation density would move from short to long separations. This effect should be more noticeable the fewer de novo complexes form after EGF binding. A screen of EGF-induced particle colocalisation events revealed that I942E-EGFR, a kinase domain mutation that allows strong EGF-induced phosphorylation[10], forms very few de novo EGF-induced complexes (Supplementary Fig. 4). FLImP data from I942E-EGFR display a decreasing ladder-like distribution of DIII–DIII separations when probed with CF640R-Affibody (Fig. 2e) and a transfer of density to longer components when probed with CF640R-EGF (Fig. 2f). These results are consistent with polymer chain architectures.

To attempt to construct a structural model of a polymer chain using long-timescale MD simulations, we searched for crystal structures of the ECM monomer in which lattice contacts might reveal previously unidentified oligomer interfaces. The structures of EGFR co-crystallised with EgA1 (PDB ID 4KRO[42]) and 9G8 (PDB ID 4KRP[42]) verified this criterion. (EgA1 and 9G8 are nanobodies (NBs) that bind DIII and stabilise the DII–DIV tether[42].) Figure 3a shows the monomer contacts in 4KRP. We hypothesised that the tethered conformation and lack of two-fold symmetry of this model, reminiscent of an X-ray structure of HER2 co-crystallised with a designed ankyrin repeat protein[43], are reasonable initial points because both counteract the formation of signalling dimers.

We then carried out two MD simulations (13.8 and 20 μs) starting from the asymmetric dimer seen in the crystal packing of 4KRP, after removing 9G8-NB and adding the TM helix and the lipid bilayer (Fig. 3b). In both simulations the ECM dimer arrived at a head-to-head dimer conformation within 1.5 μs and remained in this conformation to the end. This dimer conformation is similar to the crystal dimer conformation in that it remains asymmetric and open-ended and maintains the *trans* interaction between the DIII of one protomer and the DIV of the other, but it bears a *trans* interaction between DI and DII. Moreover, the tethered conformation of the crystal structure was consistently unstable, as in both simulations DI and DIII in both protomers gained stable *cis* interaction with one another, giving rise to a conformation similar to that in the proposed back-to-back inactive dimer[23] in terms of DI, DII, and DIII. Intriguingly, the ECMs of neither of the two protomers in the head-to-head conformation bear close contact with the membrane. Encouragingly, the bulk of the amino acids involved in the head-to-head interface (6–238) is missing in the constitutively active variant EGFRvIII prevalent in glioblastoma (6–273)[44]. If amino acids 6–273 are deleted post-simulation in the model, the monomers break apart (Fig. 3c), suggesting biological relevance.

We next added the 9G8-NBs to the starting ECM dimer and similarly simulated a 9G8-bound head-to-head dimer up to 20 μs, starting from the conformation seen in PDB 4KRP with the TM helices and the lipid bilayer added (Fig. 3d). Consistent with the two simulations of the "naked" dimer where 9G8-NB was removed, the asymmetry and the open-endedness of the dimer remained, the DIII–DIV *trans* interaction of the ECM dimer was

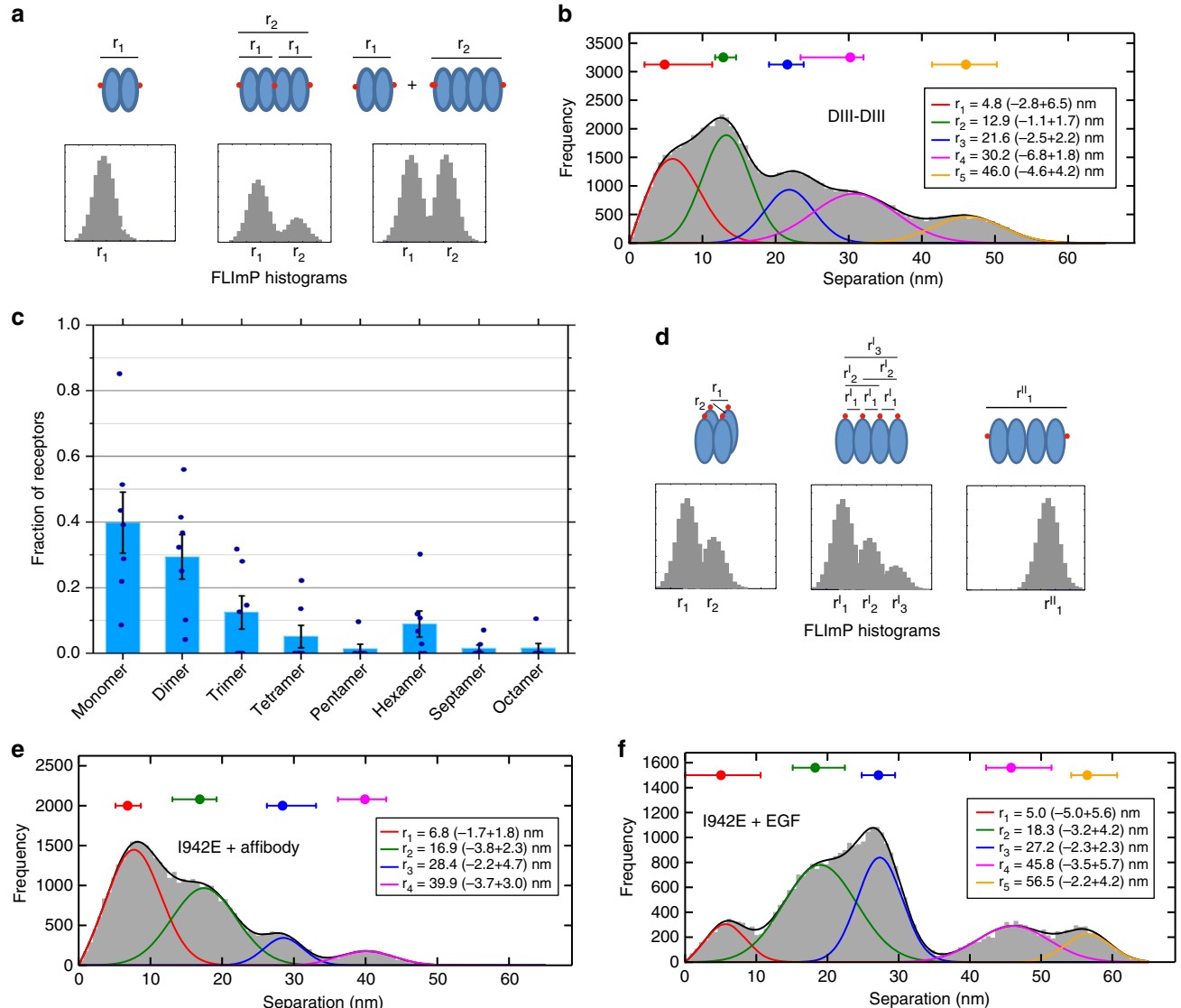

**Fig. 2** FLImP measurement of pairwise DIII-binding Affibody separations. **a** Cartoon of FLImP histograms. Left: A dimer (receptors, blue; label, red) gives rise to one separation, the empirical posterior of which has a confidence interval (CI) that depends on the combined localisation errors of the two molecules[28]. CI size determines resolution. CIs less than the required resolution are retained according to signal-to-noise without bias[12,28–30,36], generating a FLImP distribution (grey) that is fitted by the sum of a number of Rician peaks[12]. If oligomers are present and internally probed (middle) and/or the distribution of species is inhomogeneous (right), the histogram contains multiple components[12]. **b** FLImP distribution (grey) of DIII–DIII separations between CF640R-Affibody molecules bound to wtEGFR on CHO cells, compiled from 68 FLImP measurements (CI ≤ 7 nm), decomposed into a sum of five components (coloured traces). The inset shows positions and error estimates. Additional statistics in Supplementary Fig. 1. The 4 nM concentration of CF640R-Affibody used labels ~20% of receptors (Supplementary Fig. 2) ensuring single particle detection[29]. FLImP is stochastic, thus independent of the CF640R-Affibody/receptor ratio if sufficient data are collected, and uses fixed cells to avoid relative receptor movements during measurements. However, systematic studies failed to find significant artifacts[12,36]. **c** Molecular-normalised fraction of receptors in oligomer species on wtEGFR-expressing CHO cells treated with 100 nM Alexa 488-Affibody, determined by pbICS[38]. Results are the mean of seven replicates. Error bars show the SD. For more details see Supplementary Fig. 3. **d** Cartoon showing expected FLImP distributions for a cyclic tetramer labelled with a 1:1 probe binder (like the Affibody) (left), a polymer chain labelled with a 1:1 probe binder (middle), and a polymer showing a 2N:2 labelling stoichiometry (like EGF)[12] (N = receptor number) (right). **e** FLImP distribution (grey) and peak decomposition of DIII–DIII separations reported by CF640R-Affibody molecules bound to I942E-EGFR on the surface of CHO cells, compiled from 36 FLImP measurements (CI ≤ 7 nm), decomposed into a sum of four peak components (coloured traces). Positions and error estimates are shown in the inset. **f** As **e** using 4 nM CF640R-EGF as a probe, compiled from 31 FLImP measurements (CI ≤ 7 nm)

stable, and the DI–DII *trans* interaction was developed. The DI–DIII *cis* interaction again developed in both protomers, reproducing the previously reported monomer conformation[23].

**Head-to-head interactions assemble curved polymer chains.** The head-to-head dimer can be extended into a higher-order

oligomer by incorporating additional protomers and repeating the head-to-head interaction (Fig. 4a). pbICS data reveal that ~25% of species are dimers, ~10% trimers, and ~5% tetramers and above (Fig. 4b). The fraction of each species as a function of oligomer size looks approximately exponential consistent with a step-wise polymerisation process. Curvature of the polymer chain in the plane of the membrane would arise because the DIII–DIII

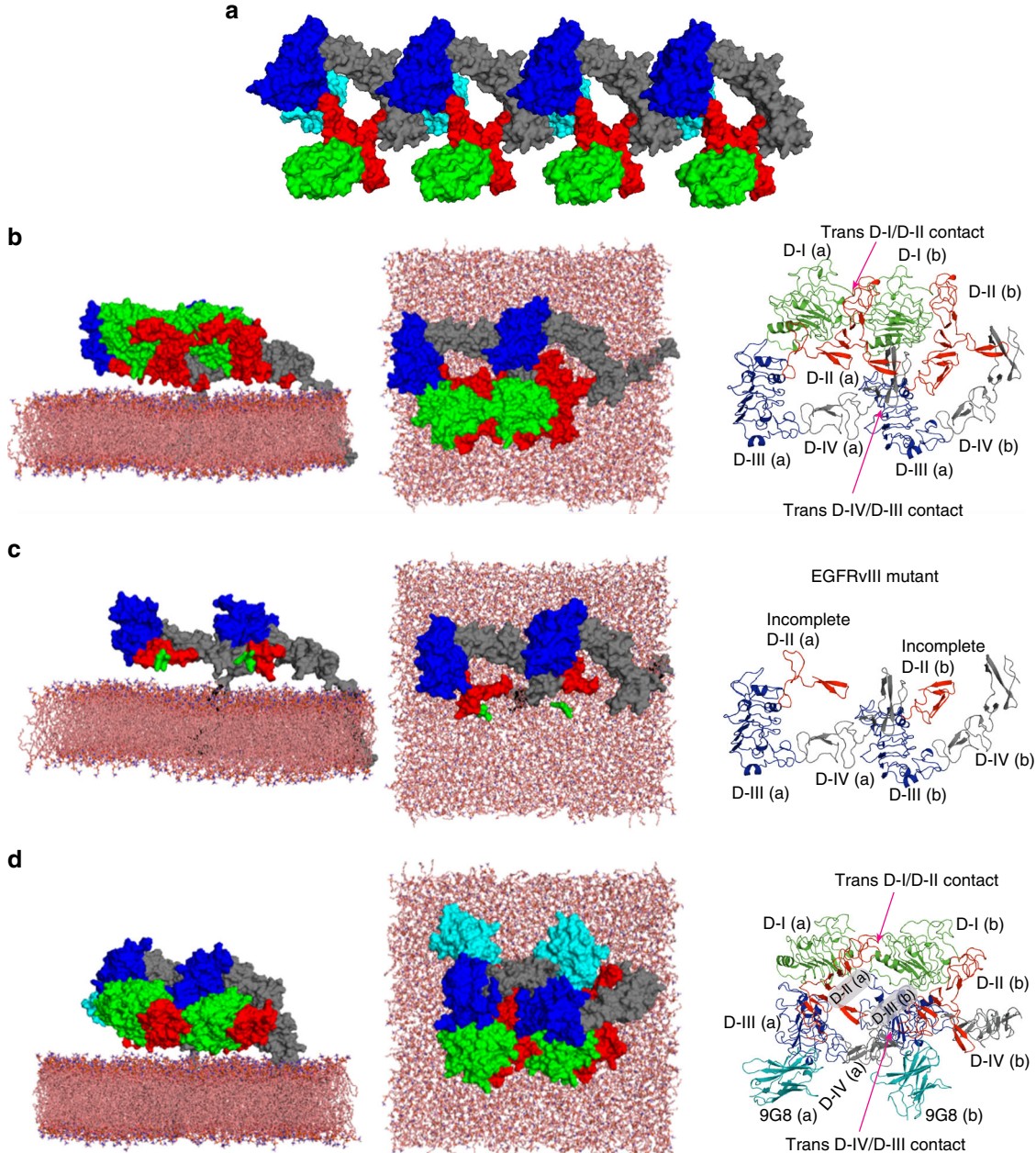

**Fig. 3** The structural model of ligand-free head-to-head EGFR dimers. **a** An open-ended oligomer model of 9G8-bound EGFR extracellular domains in the inactive conformation built using the crystal contacts in the monomer structure in PDB ID 4KRP[42], where 9G8-NB is coloured in cyan, EGFR DI in green, DII (red), DIII (blue), and DIV (grey). **b** A simulation-generated dimer structure of free EGFR extracellular domains and their TM domains in the lipid bilayer. The simulation was started from the crystal dimer of 9G8-bound EGFR extracellular domains in the tethered conformation in which the two copies of the 9G8-NB were removed from the simulation system. The images are based on the snapshot of the simulation at 20 μs. One of the two transmembrane helices is visible (left and middle panels). The right panel shows the dimer viewed from the membrane, where the *trans* interaction between DI and DII and the interaction between DIV and DIII are highlighted; the domains of monomer a and b are labelled. **c** By deleting residues 6–273 from the dimer structure in **b**, the dimer structure directly mapped to the EGFRvIII glioblastoma mutant. As shown, this dimer structure is likely not viable in EGFRvIII as the *trans* interaction between DI and DII is precluded by the deletion. **d** A simulation-generated dimer structure of 9G8-bound EGFR extracellular domains starting from a crystal dimer of 9G8-bound EGFR extracellular domains in the tethered conformation. These images are based on the snapshot of the simulation at 20 μs. Invisible from this image are the TM helices embedded in the membrane. The right panel shows the 9G8-bound dimer viewed from the membrane, where the trans-dimer interaction between DI and DII, and between DIV and DIII are highlighted

separation between nearest neighbours is larger than DI–DI (Fig. 3b). One can imagine such curved geometry would facilitate receptor packing around the mouths of vesicles like, for example, caveolae, which have a ~50 nm diameter[45]. The predicted DIII–DIII separations (Fig. 4a, blue) are encouragingly consistent with the experimentally measured separations (Fig. 2b; Supplementary Fig. 1). Moreover, the largest separation resolved (46 nm) corresponds to the prediction for the largest octamer species detected by pbICS.

To further assess the possibility that this model represents the architecture of non-monomer complexes, we conjectured that ΔC-EGFR, which mirrors the amino acid sequence used in the

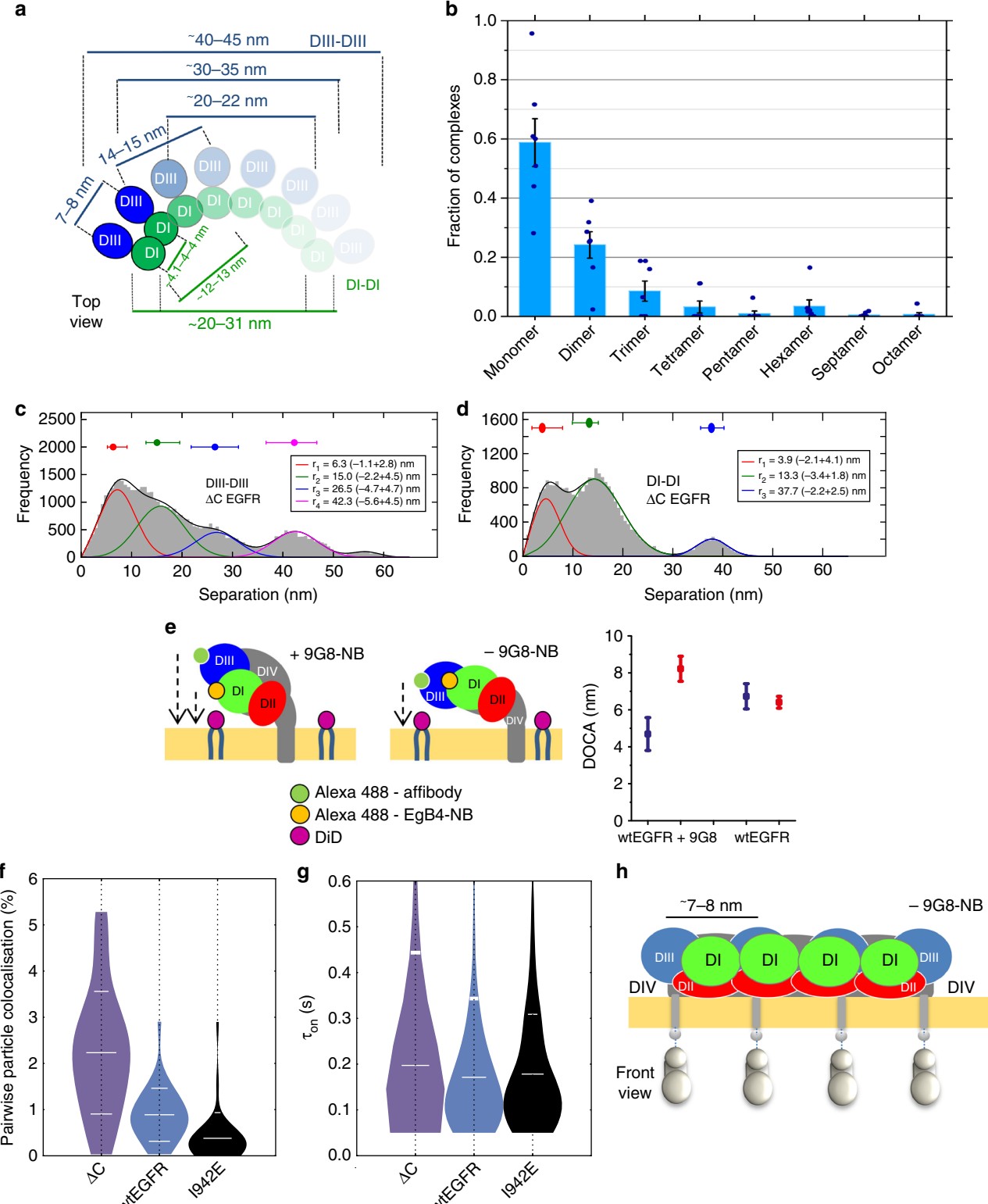

simulated head-to-head dimer model (no ICM), should form polymers consistent with the model. To test this, we probed DIII–DIII separations on CHO cells that express ΔC-EGFR. The separations resolved are consistent with the head-to-head polymerisation model predictions (Fig. 4c, Supplementary Fig. 7).

The model also predicts shorter DI–DI separations (Fig. 4a, green). We tested this using a CF640R derivative of an anti-EGFR EgB4 nanobody (EgB4-NB) that specifically binds DI of EGFR

without inducing activation[42,46] (Supplementary Fig. 6). The distribution of DI–DI separations is significantly compressed towards shorter values (Fig. 4d, Supplementary Figs. 8 and 9a–c). Being smaller, fewer DI–DI separations are resolved, and those resolved are consistent with model predictions.

Predicted distances from DI and DIII to the membrane were tested by FRET[32] (Fig. 4e). In parallel experiments we labelled DI or DIII of wtEGFR with Alexa 488 derivatives of EgB4-NB or

**Fig. 4** The architecture of ligand-free head-to-head polymers. **a** Polymer chain formed by repeating the head-to-head interface based on separations in Supplementary Fig. 5a, b and Supplementary Tables 1 and 2 (DII and DIV excluded for simplicity). The intensity is graded according to pbICS results in **b**, which show the data in Fig. 2c re-normalised to reveal the fractions of oligomer species on wtEGFR-expressing CHO cells treated with 100 nM Alexa 488-Affibody. Results are the mean of seven replicates. Error bars show SD. **c** FLImP distribution (grey) of DIII–DIII separations in ΔC-EGFR-expressing CHO cells treated with 4 nM CF640R-Affibody, from 41 FLImP measurements (CI ≤ 6 nm). The inset shows positions and error estimates (additional statistics in Supplementary Fig. 7). **d** As **c** but from ΔC-EGFR-expressing cells treated with 8 nM CF640R-EgB4-NB (DI–DI separations), from 32 FLImP measurements (CI ≤ 8 nm). Differences with the DIII–DIII distribution are significant (Supplementary Fig. 8). **e** Left and centre: Cartoons showing a side view of DI and DIII separations from the membrane in head-to-head complexes in the presence and absence of bound 9G8-NB based on Supplementary Fig. 5c–f and Supplementary Table 3. Right: FRET-derived separations from the membrane-DiI to DI (Alexa 488-EgB4-NB, blue) or DIII (Alexa 488-Affibody, red). The bar chart was derived from the measurements in Supplementary Fig. 10. (As predicted by the model, EGFR also forms oligomers on cells treated with 200 nM 9G8-NB (Supplementary Fig. 11a, b). **f** Two-colour SPT on live cells at 37 °C showing the fraction of tracks where two particles labelled with Alexa 488-Affibody and CF640R-Affibody spent ≥5 frames (250 ms) together within <1 pixel (pairwise particle colocalisation fraction) for cells expressing ΔC-EGFR, wtEGFR and I942E-EGFR. Horizontal spreads separate data points (~5000) within each condition. **g** Distribution of the duration of pairwise interactions ($\tau_{ON}$) in **f**. Horizontal lines mark mean and SD. Coincidental colocalisation statistics were accounted for[12]. *p*-Values (two-tailed Kolmogorov–Smirnov test) are in Supplementary Fig. 12. **h** Front view of a ligand-free oligomer illustrating the separation between non-interacting ICM units predicted by extracellular head-to-head interactions. All panels: DI, green; DII, red; DIII, blue; DIV, dark grey; plasma membrane, yellow; TM, grey; TKD silver

Affibody (FRET-donors) in the presence and absence of bound 9G8-NB. The plasma membrane was treated with dialkylcarbocyanine probe DiIC$_{18}$(5) (DiI) (FRET-acceptor). As shown in Fig. 4e, DI is on average closer to the bilayer than DIII when 9G8-NB is bound, while in its absence DI and DIII separations are similar. The FRET measurement reports the average of monomers and non-monomers; however, because these are predicted to have similar extracellular conformations[23], these FRET results are consistent with the model predictions. In conclusion, our FLImP and FRET data strongly favour the head-to-head interface as the building block of ligand-free polymer chains.

**ICMs in head-to-head polymers do not form dimers**. We reasoned that extracellular and intracellular interactions should together assemble more stable complexes than those assembled by extracellular interactions alone. Given this, the contribution of the ICMs to oligomerisation could be ascertained by comparing the stability of the complexes formed by ΔC-EGFR (no ICM) and the full-length wtEGFR and I942E-EGFR. This comparison is possible because ΔC-EGFR, wtEGFR, and I942E-EGFR form indistinguishable polymer chains (Figs. 2b, e, 4c, Supplementary Fig. 9d–f). For this investigation we used two-colour SPT, a method previously used to report the incidence of pairwise receptor particle interactions (colocalisation fraction) and their duration ($\tau_{ON}$)[12,14]. Receptors were labelled with a 1:1 mixture of Alexa 488-Affibody and CF640R-Affibody. Results show that ΔC-EGFR forms the most complexes (Fig. 4f), which have an indistinguishable duration to those formed by wtEGFR and I942E-EGFR (Fig. 4g). These results argue against a significant contribution from intracellular interactions to oligomerisation (Fig. 4h).

**The aTKD dimer breaks head-to-head oligomers**. The sizeable separation between the N-termini of the TM helices in the head-to-head dimer predicts that the stabilisation of the aTKD dimer, by enforcing the N-crossing TM dimer[40], would break the head-to-head interaction. To test this, we stabilised the aTKD dimer by treating wtEGFR-expressing CHO cells with erlotinib[26], a type I TKI[47]. The four DIII–DIII separation components resolved (Fig. 5a) are at positions consistent with four of the five resolved in the absence of erlotinib (Fig. 2b). We also determined the proportion of measurements consistent with each of the mean positions, an analysis that revealed a dominant 5 nm component and a significant decrease in separation density at longer positions when compared with naked wtEGFR (Fig. 5b). This result correlates aTKD dimer formation with oligomer loss.

To examine if oligomers split when the aTKD dimer is stabilised in the absence of TKI, we treated CHO cells expressing wtEGFR with methyl-β-cyclodextrin (MβCD), which depletes cholesterol from the plasma membrane and activates EGFR in a ligand-independent manner[48] (Fig. 5c). The associated increase in aTKD dimers returned results remarkably similar to those found upon erlotinib treatment, namely DIII–DIII separations with a dominant peak at ~4 nm and significantly fewer oligomers (Fig. 5b, d). Together these results support the notion that the stabilisation of the aTKD dimer, either in the presence or absence of erlotinib, disrupts oligomer formation.

**Kinase-mediated stalk-to-stalk and back-to-back dimers**. Besides disrupting oligomers, the dominance of one DIII–DIII separation component in the FLImP distributions of Fig. 5a, d indicates that the stabilisation of the aTKD dimer promotes an ECM dimer. This is supported by the increased receptor–receptor colocalisation following erlotinib binding, as shown by SPT (*p* < 0.01, Kolmogorov–Smirnov test, two-tailed) (Fig. 5e). The formation of erlotinib-bound aTKD dimers is also consistent with the increased duration of the interactions between receptors when compared with naked wtEGFR where only extracellular contacts are present (*p* < 0.01, Kolmogorov–Smirnov test, two-tailed) (Fig. 5f). The resolution of FLImP (~5–7 nm) is, however, insufficient to distinguish between the head-to-head dimer (~7 nm DIII–DIII separation) and the ECM dimer induced by erlotinib (~5 nm DIII–DIII separation). We nevertheless reasoned that these dimers are likely to be different because N-crossing TM dimers would disrupt head-to-head interfaces.

To investigate the ECM dimer linked to the aTKD dimer, we considered the stalk-to-stalk structure detected by EM[26]. This dimer is upright, not-extended, and held by close C-terminal DIV–DIV contacts that afford linking with the N-crossing TM dimer and conformational flexibility (Fig. 1e). To investigate the presence of DIV–DIV contacts, we examined IIIV/KKRE-EGFR, in which mutations I545K, I556K, I562R, and V592E (IIIV/KKRE-EGFR) replace a hydrophophic surface of DIV with charged amino acids[11] (Fig. 5g). As shown in Fig. 5h, the IIIV/KKRE-EGFR mutations should not disrupt head-to-head interactions. Consistent with this, FLImP showed IIIV/KKRE-EGFR forms as many head-to-head dimers and oligomers as wtEGFR (Fig. 5b, i). The IIIV/KKRE-EGFR mutations are however poised to disrupt DIV–DIV contacts. Thus, if formation of the aTKD dimer is facilitated by DIV–DIV contacts in the ECM, then cholesterol depletion should induce less phosphorylation in IIIV/KKRE-EGFR. The absence of increased phosphorylation in cholesterol-depleted IIIV/KKRE-EGFR-expressing cells is

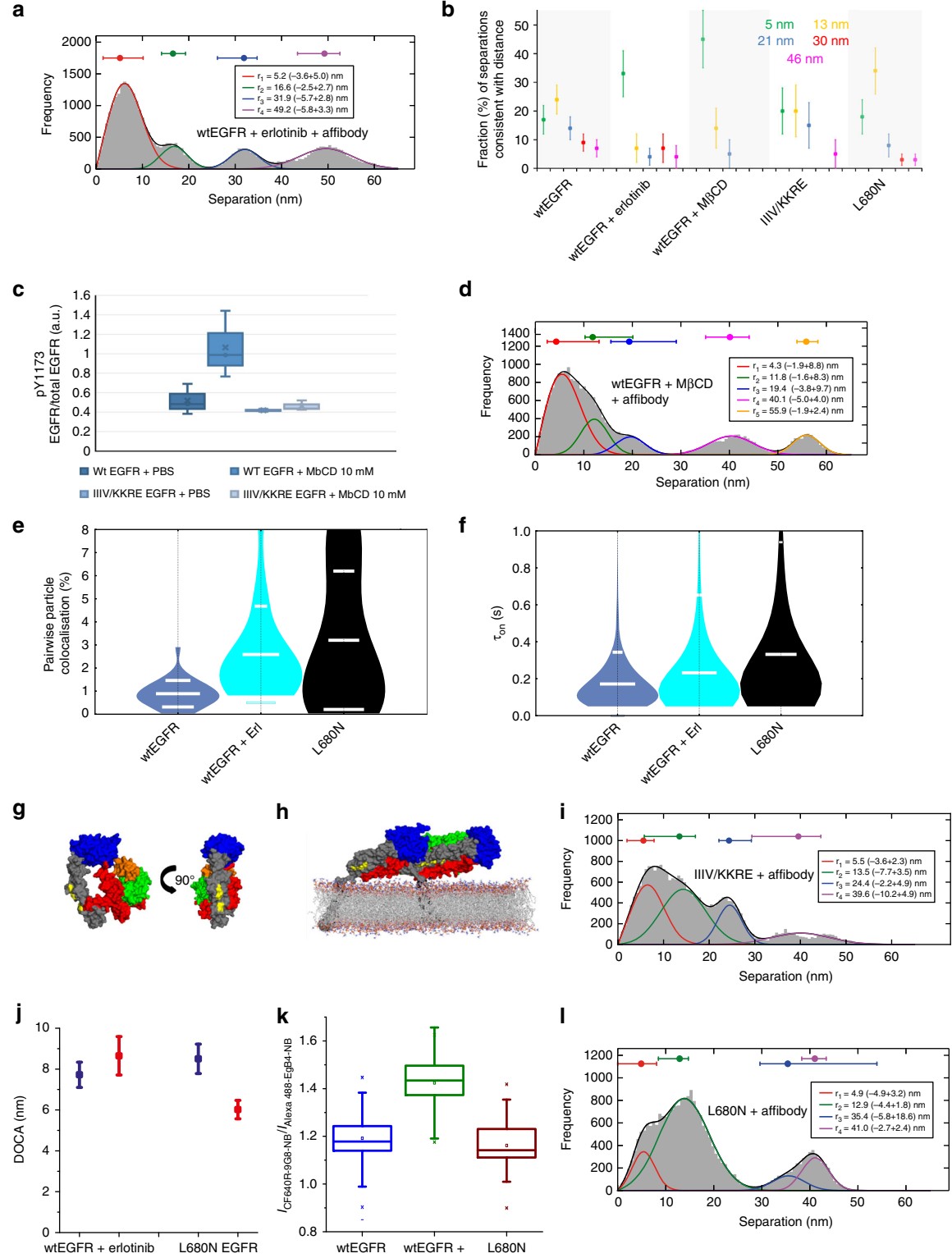

consistent with the requirement for DIV–DIV contacts for stabilisation of a functional aTKD dimer (Fig. 5c).

We next used FRET to measure the separations from DI and DIII to the membrane in erlotinib-bound wtEGFR. Assuming an upright orientation, the similar DI- and DIII-membrane separations found are consistent with a non-extended conformation, as that predicted by the stalk-to-stalk dimer (Fig. 5j). In turn, error analysis of the FRET data confirms a conformation different from

that of naked wtEGFR (Supplementary Fig. 10a, c, Supplementary Table 4). To investigate conformational flexibility, we compared the binding of 9G8-NB to erlotinib-bound wtEGFR and naked wtEGFR. We found that 9G8-NB, which stabilises the tether[42], displays greater binding to erlotinib-bound wtEGFR (Fig. 5k), consistent with a greater ability to adopt a tethered conformation. This would be afforded by the increased flexibility of the stalk-to-stalk dimer.

**Fig. 5** Kinase-mediated ligand-free dimers adopt two ECM dimer architectures. **a** FLImP distribution (grey) of DIII–DIII separations between CF640R-Affibody molecules bound to wtEGFR on CHO cells treated with 1 μM erlotinib, compiled from 29 FLImP measurements (CI ≤ 6 nm), decomposed into a sum of four components. The concentration of CF640R-Affibody was 4 nM. **b** Number of measurements consistent with the mean distances resolved in the FLImP distribution of wtEGFR (Fig. 2b) (associated FLImP distributions in Supplementary Fig. 13). Errors were assesed with bootstrap-resampling[12]. **c** wtEGFR and IIIV/KKRE-EGFR phosphorylation in Y1173 on CHO cells treated or untreated with 10 mM MβCD. Box plots show inclusive median as a line, 25th and 75th quartile as edges, calculated on $n = 3$ repeats (representative western blots in Supplementary Fig. 14). **d** As **a** treated with 10 mM MβCD, from 20 FLImP measurements (CI ≤ 7 nm). **e** Pairwise particle colocalisation fraction from two-colour SPT on live cells at 37 °C. **f** Duration of pairwise interactions ($\tau_{ON}$) in **e**. Horizontal spreads separate data points (~5000) within each condition. **g** Crystal structure of tethered wtEGFR (PDB ID 1NQL[5]) highlighting the location of I545K, I556K, I562R, and V592E (yellow). Colours: DI (green), DII (red), DIII (blue), DIV (grey), EGF (orange). **h** Head-to-head dimer highlighting the residues mutated in the IIIV/KKRE mutant (yellow). **i** As **a** but for the IIIV/KKRE-EGFR mutant from 22 FLImP measurements with CI ≤ 7.5 nm. **j** FRET-DOCA from DI (blue) and DIII (red) to the membrane for wtEGFR + erlotinib, and L680N-EGFR, derived from measurements in Supplementary Fig. 10. FRET probes as in Fig. 4e. **k** Ratio between CF640R-9G8-NB and Alexa 488-EgB4-NB binding after chemical fixation. wtEGFR, blue; wtEGFR + erlotinib, green; L680N-EGFR, red. Line, median; box edges, 25th and 75th quartile, crosses 5th and 95th quartile, calculated over 30 repeats. Example images and analysis are in Supplementary Fig. 15. **l** As **a** but for L680N-EGFR-expressing cells, from 20 FLImP measurements (CI ≤ 6 nm). Lower resolution (8 nm) versions of **a**, **d**, and **l** with ~2-fold more FLImP measurements show the profile of the distributions is conserved (Supplementary Fig. 16)

We next reasoned that, if the stalk-to-stalk and aTKD dimers are not just linked across the membrane, but conformationally coupled, the stabilisation of the sTKD dimer[6,10,22] might promote an ECM dimer different to the stalk-to-stalk dimer, and possibly similar to the ligand-free back-to-back architecture proposed by MD simulations[23] (Fig. 1d). To test this possibility, we examined L680N-EGFR, in which a mutation in the N-lobe of the kinase domain hinders the aTKD dimer interface[10]. L680N is analogous to I682Q, which gave rise to the 5CNN[6] crystallographic sTKD structure. As shown in Fig. 5l, the dominance of a different DIII–DIII separation in L680N-EGFR (in this case 13 nm) suggests that the L680N mutation promotes inside-out an extracellular dimer different to the stalk-to-stalk dimer. Further evidence for L680N-induced dimerisation is provided by the increased number and duration of the interactions holding L680N-EGFR dimers together ($p < 0.01$, Kolmogorov–Smirnov test, two-tailed) (Fig. 5e, f). The distribution of DIII–DIII separations also displays fewer separations (>20 nm) than in wtEGFR (Fig. 5b), suggesting that formation of the sTKD dimer is incompatible with the head-to-head interface.

FRET results from L680N-EGFR showed a greater separation from DI to the membrane than from DIII (Fig. 5j). These results argue that the dimers formed by L680N-EGFR are upright and adopt an extended conformation[49]. Given that the dimension of the Affibody is 3.7 nm[37], the 13 nm DIII–DIII separation is within range of the predictions of the ligand-free back-to-back dimer, in which the distance between binding sites is ~7 nm. In turn, the lower binding of 9G8-NB to L680N-EGFR than to wtEGFR bound to erlotinib (Fig. 5k) is consistent with the smaller conformational flexibility expected in the back-to-back dimer. The data therefore argue that the aTKD (sTKD) dimer is structurally coupled across the plasma with the previously proposed stalk-to-stalk (back-to-back) dimers.

**Inside-out signals regulate cancer mutant dimer geometry**. To investigate changes in ligand-free EGFR structure accompanying dysregulated signalling, we examined L834R-EGFR and T766M-EGFR (or L858R and T790M in numbering that include the 24 amino acids signal peptide). These two mutations are prevalent in non-small cell lung cancer (NSCLC) and located within exons 18–21 in the vicinity of the ATP-binding site (Fig. 6a). L834R-mutated NSCLC tumours respond to first-generation TKIs (like erlotinib)[50]. T766M increases the affinity for ATP and confers resistance to first- and second-generation TKIs[51]. Western blots confirmed the constitutively phosphorylated status of these mutants (Supplementary Fig. 17). We collected FLImP data and compared the results with wtEGFR, ΔC-EGFR, wtEGFR +

erlotinib and wtEGFR + MβCD, and L680N-EGFR (Fig. 6b–e). As positive control for kinase-mediated dimerisation we used Δ973-EGFR, in which truncation of the C-terminus abolishes the autoinhibitory interaction with its own kinase[6]. As negative control we used Δ698-EGFR, which lacks the kinase domain.

From the FLImP distributions we quantified the DIII–DIII separations consistent with 5 nm (green), 13 nm (yellow), <15 nm (pink) and >20 nm (blue) (Figs. 6b–e). Consistent with results above (Fig. 5), we found that oligomers (>20 nm) are reliably more abundant when one would not expect kinase-mediated dimerisation (Fig. 6b). Indeed, Fig. 6c shows the FLImP distributions of the three conditions displaying more oligomers. Unsurprisingly, with wtEGFR (low phosphorylation) and ΔC-EGFR (no ICM), we found Δ698-EGFR (no TKD). In contrast, oligomers were consistently depleted in conditions where one would expect kinase-mediated dimerisation. Among these conditions, Fig. 6d shows the distributions for wtEGFR upon erlotinib or MβCD treatments, in which the formation of the aTKD dimer promotes a dominant ~5 nm component that reports the formation of stalk-to-stalk ECM dimer at the expense of head-to-head interfaces, as shown in Fig. 5. In this group, we also found L834R-EGFR, in which the L834R mutation stabilises the aTKD dimer interface by suppressing the intrinsic disorder in the N-lobe kinase dimerisation interface[52]. According to conformational coupling, the dominant 5 nm FLImP peak and fewer oligomers found in L834R-EGFR would suggest that the stabilisation of the aTKD dimer also promotes the stalk-to-stalk dimer in this cancer mutant. We used FRET to validate this notion (Supplementary Fig. 10e). (Later we also discuss the similarity of the results for L834R-EGFR and wtEGFR under erlotinib and MβCD treatments with additional data (Fig. 7).) Lastly, Fig. 6e shows distributions with a sizeable 13 nm component and a depleted oligomer population. Consistent with conformational coupling, inhibition of the aTKD dimer in L680N-EGFR is reflected by the relatively lower 5 nm component in this group. In contrast, the sizes of the 5 and 13 nm peaks, resolved by BIC analysis (Supplementary Fig. 11c, d), are comparable in T766M-EGFR and Δ973-EGFR, whose *trans* activation depends on the formation of the aTKD dimer[51]. We were reassured to find that deletion of the C-terminus (Δ973-EGFR) resulted in oligomer loss and the promotion of dimers (Fig. 6b–e). This is consistent with the role of the C-terminus in preventing kinase-mediated dimerisation[53].

**T766M-EGFR can form symmetric tyrosine kinase dimers**. We were surprised to find significant numbers of 13 nm separations in the constitutively phosphorylated T766M-EGFR (Fig. 6e). If, as

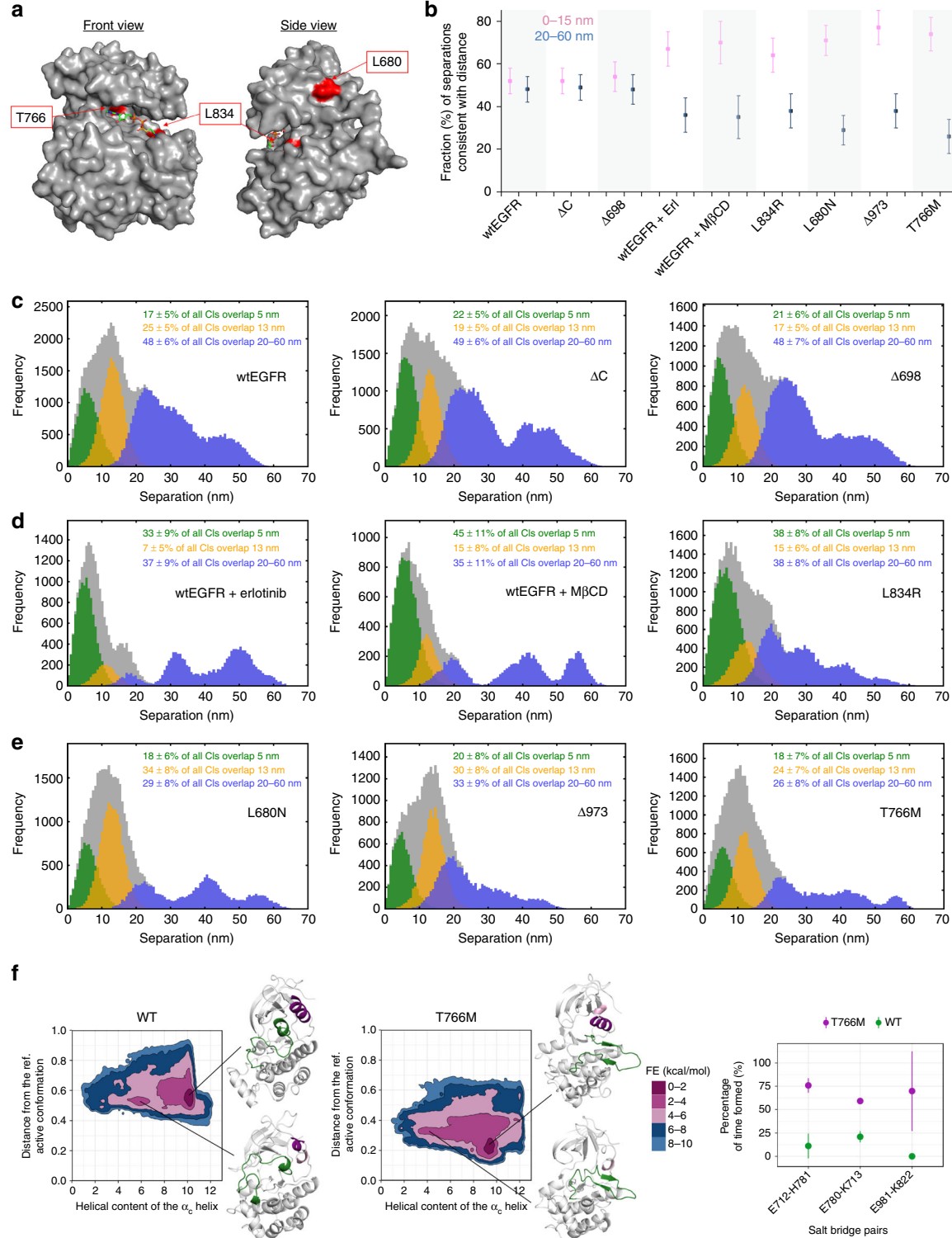

shown by FLImP and FRET results for L680N-EGFR, this peak reports extended back-to-back dimers, and, as proposed by MD simulations[23], this dimer is structurally coupled to the sTKD dimer, then, counterintuitively, our results predict that T766M-EGFR forms significant numbers of sTKD dimers. To investigate this unexpected prediction, we carried out MD simulations of the sTKD dimer and reanalysed fully converged free-energy landscapes from parallel tempering metadynamics (PTmetaD) simulations, in the well-tempered ensemble for wtEGFR and T766M-EGFR, as in Sutto et al.[34].

We reweighted the MD trajectories and projected the free-energy surface (FES) onto the helical content of the αC helix and the contact map corresponding to the active (open) state. As expected, the FES for wtEGFR shows a single global minimum that corresponds to the intact αC helix with the A-loop in the Src-like inactive conformation (Fig. 6f, left). There is also a higher-energy minimum populated with structures with a disordered αC helix. In the case of T766M mutant, this minimum is broader and shifted towards lower values that correspond to higher disorder of the αC helix (Fig. 6f, middle). The global minimum with an intact

**Fig. 6** Kinase-mediated regulation of ECM dimer geometry in cancer-associated intracellular mutants. **a** Kinase domain structure showing the positions of relevant amino acids. **b** Fractions of FLImP measurements whose ranges of 69% confidence overlap with the ranges of DIII–DIII separations expected for dimers (0–15 nm) (pink) or oligomers (20–60 nm) (blue) collected from cells expressing the receptor mutants and/or under the conditions noted in the X-axis. The associated FLImP distributions are in Supplementary Fig. 18. **c–e** The FLImP distributions (grey) and the distributions compiled from the FLImP measurements whose 69% CIs overlap with DIII–DIII separations = 5 nm (green), 13 nm (yellow), or >20 nm (blue) collected from cells expressing wtEGFR, ΔC-EGFR, Δ698-EGFR, wtEGFR on cells treated with 1 μM erlotinib, wtEGFR on cells treated with 10 mM MβCD, L834R-EGFR, L680N-EGFR, Δ973-EGFR and T766M-EGFR. The insets show the fraction of separations consistent with each distance. Errors were assesed with bootstrap-resampling[12]. **f** Free-energy surfaces as a function of the helical content of the αC helix and the distance from the reference active conformation (a contact map corresponding to the active extended A-loop conformation) as obtained from PTmetaD simulations of the wtEGFR and T766M-EGFR mutant. The central structures of the most populated clusters are shown (left and middle). The A-loop is coloured green, while the αC helix is shown in purple if it forms an α-helix and in pink if it forms a $3_{10}$ helix. Salt-bridge interactions formed at the dimer interface in the last μs of the unbiased MD simulations of the symmetric dimers (right panel). The mean values and the standard deviations are calculated across monomers. The salt bridge was considered to be formed if the minimal distance between the side chains of residues in question was <4 Å. (More information on the calculations can be found in Supplementary Methods)

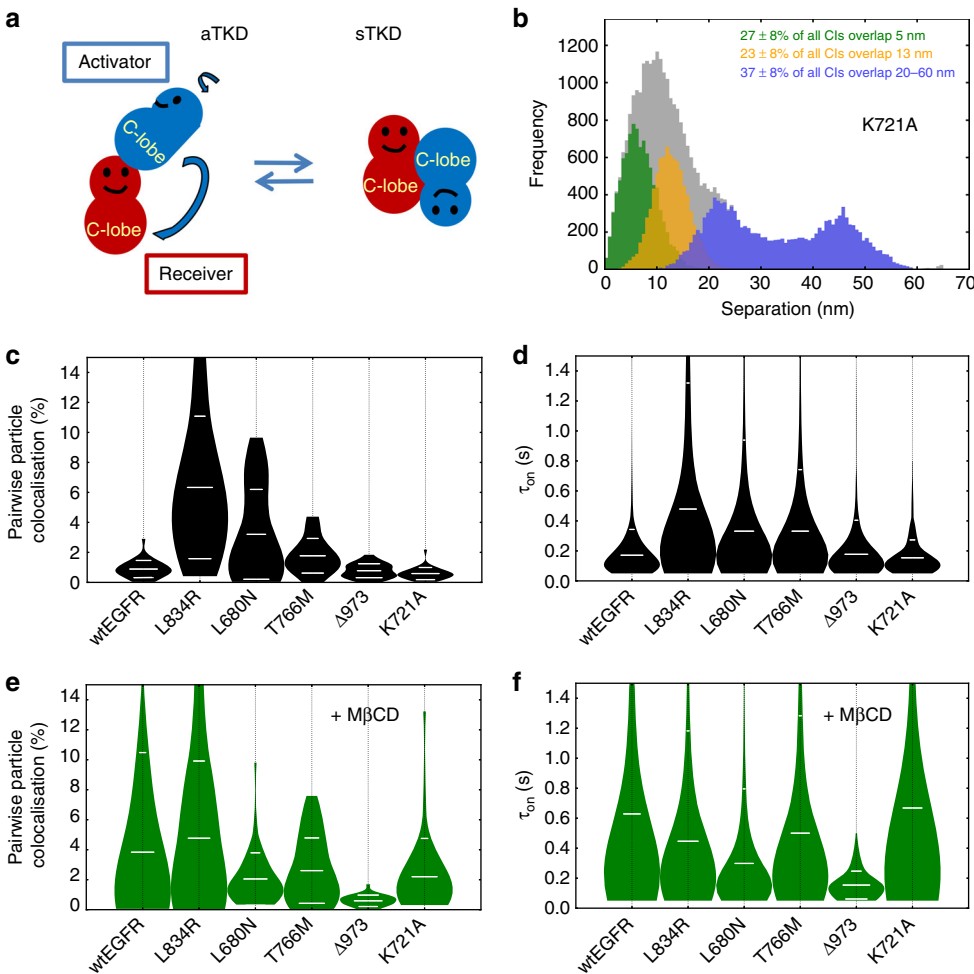

**Fig. 7** Equilibrium between the aTKD and sTKD dimers. **a** Cartoon of the two dimer configurations of the kinase domain. Starting from the aTKD dimer, a counter clockwise rotation of the activator (blue) along its vertical axis followed by a clockwise rotation along an axis perpendicular to the plane will allow the sTKD to form. **b** The FLImP distribution (grey) and the distributions compiled from the FLImP measurements whose 69% CIs overlap with DIII–DIII separations = 5 nm (green), 13 nm (yellow), or >20 nm (blue) collected from cells expressing K721A-EGFR compiled from 29 FLImP measurements with CI ≤ 7 nm. The inset shows the fraction of separations consistent with each distance. Errors were assesed with bootstrap-resampling[12]. **c** Fraction of tracks where two different-colour particles labelled with Alexa 488-Affibody and CF640-Affibody spent ≥5 frames (250 ms) together within <1 pixel (denoted fraction of pairwise particle colocalisation fraction) and **d** distribution of the duration of these pairwise interactions ($\tau_{ON}$). Horizontal spreads separate data points (~5000) within each condition. Horizontal white lines mark the mean and SD. Coincidental colocalisation statistics were accounted for. **e**, **f** As **c**, **d** but in the presence of 10 mM MβCD. p-Values of the significance of differences between conditions are in Supplementary Fig. 20

αC helix is also present and both minima are populated with structures with an active A-loop conformation.

These results indicate that the T766M mutation has two effects. On the one hand, it is clear from the converged FES of the monomer that it destabilises the aTKD dimer interface by affecting the stability of the αC helix. On the other, the long MD simulations of the sTKD dimer in wtEGFR and T766M-EGFR show more stability for the single mutant (as indicated by root-mean-square fluctuations, Supplementary Fig. 19a) and suggest that T766M stabilises the sTKD dimer through a network of interactions (Fig. 6f (right), Supplementary Fig. 19b). In particular, as shown in Supplementary Fig. 19c, we see that Q767 engages in interactions with the interface residues (such as E981) located below the AP-2 helix that refolds in this dimer (it is disordered in the monomer). Further evidence to this observation comes from Jura et al.[22] where it is shown the E981R mutation facilitates EGFR phosphorylation. This implies that the inactive sTKD cannot form.

**The aTKD/sTKD dimer equilibrium depends on the C-terminus.** The finding of significant numbers of sTKD dimers in the constitutively active T766M-EGFR, but not in wtEGFR (Fig. 6f), argues against a main autoinhibitory role for the sTKD dimer[25]. Alternatively, in the absence of structural impediments, kinase-mediated dimerisation might inherently give rise to both active aTKD and inactive sTKD dimer configurations (Fig. 7a). Recently, solid state NMR showed that EGFR structure is dynamic[54], suggesting that different conformations may exchange in time. This is supported here by three observations: firstly, in L834R-EGFR, where the aTKD dimer interface is super-stable[52], there are few sTKD dimers (Fig. 6d); secondly, in L680N-EGFR, in which the aTKD dimer is inhibited, the sTKD dimer population is boosted (Fig. 6e); and thirdly, in the absence of constraints, comparable numbers of aTKD and sTKD dimers form (like in T766M-EGFR) (Fig. 6e). Indeed, K721A-EGFR, in which the lysine 721 to alanine mutation in the ATP-binding pocket renders the kinase dead[55], is another example in which both aTKD and sTKD dimers form at the expense of oligomers (Fig. 7b). This result highlights the critical role played by the kinase core in regulating distal dimerisation interfaces[56] and the challenges associated to intervening with TKIs.

To test the equilibrium hypothesis we examined the effect of mutations on the kinetics of particle interactions. Crucially, we can interpret SPT results knowing the stoichiometry and architecture of the different conditions (Figs. 2–6). Results reveal that the kinase-mediated dimers induced by the L834R, L680N, and T766M mutations are more abundant (Fig. 7c) and longer lived (Fig. 7d) than the extracellular head-to-head interfaces formed by wtEGFR ($p < 0.01$). T766M-EGFR and L680N-EGFR dimers are similarly stable (Fig. 7c), underscoring the preference of T766M-EGFR for the sTKD dimer.

Also shown in Fig. 7c, d, the dimers induced by C-terminal truncation in Δ973-EGFR are fewer than those induced by the L834R, L680N, and T766M mutations. This is consistent with the pivotal role played by the C-terminus in the stabilisation of active and inactive kinase dimers[6,22]. In turn, the low stability of the dimers formed by K721A-EGFR is consistent with the key role played in stabilising dimer interfaces by the opening and closing of the ATP-binding site during ATP hydrolysis[56].

Using coarse-grained metaDynamics free-energy calculations, Lelimousin et al.[57] proposed that a change in bilayer thickness, such as that induced by cholesterol depletion[58], promotes a transition from less stable TM domain dimers, including the C-crossing configuration associated to the sTKD[23], to the most stable N-crossing TM dimer. Given this, we speculated that if

receptor populations bearing sTKD and aTKD dimers are in equilibrium, we could alter the kinetics by depleting plasma membrane cholesterol. Indeed, MβCD promotes in T766M-EGFR an increase in the number of dimers[48] (Fig. 7c, e) in which the duration of the interactions is consistent with the formation of active aTKD dimers. The latter is borne by the striking similarity between T766M-EGFR + MβCD with wtEGFR + MβCD and L834R-EGFR (Fig. 7f). Note these dimers are phosphorylated, and hence longer lived than erlotinib-bound wtEGFR dimers (Fig. 5f). As the predicted equilibrium would dictate, cholesterol depletion does not increase the number of dimers formed by L834R-EGFR, which is depleted of sTKD dimers (Fig. 6d), or L680N-EGFR, where the N-lobe mutation inhibits the aTKD dimer (Fig. 7d, f). These results are therefore not only consistent with a shift in the equilibrium from sTKD to aTKD dimer configuration, but also with the key role predicted for motion correlations between the TM, JM and kinase domains in the stabilisation of the aTKD dimer[56]. Interestingly, the shift in equilibrium from sTKD to aTKD dimer does not take place in Δ973-EGFR, but it is strongly present in K721A-EGFR, showing that the equilibrium between kinase dimer configurations depends on the C-terminus and not just on phosphorylation.

**Discussion**

In summary, the body of data reported here describes the architectures of three EGFR non-monomer ligand-free species populating the cellular basal state (Fig. 8). We propose a structural model of a polymer chain assembled through a previously undiscussed extracellular head-to-head interaction. Investigations on intracellular mutants and treatments found no evidence of intracellular interactions in the head-to-head polymers. The proposed role of the head-to-head interaction is therefore two-fold. On the one hand, it drives oligomer assembly, thereby bringing receptors together and priming them for a quick response to growth factor. On the other, it maintains TM helices sufficiently apart to oppose the formation of kinase-mediated dimers. Indeed, kinase-mediated dimerisation breaks the head-to-head interaction, suggesting the latter is important for autoinhibition.

The curved architecture of head-to-head polymers provides an elegant solution to the autoinhibition conundrum. Achieved via homo-extracellular interactions, such curved architecture delivers the largest possible consistent separation between TM helices in multimers, while assembling polymers commensurate with the diameter around relevant plasma membrane vesicles[45]. The head-to-head interaction averts the two-fold symmetry advantageous to kinase-mediated interactions, and, by holding receptors via DI, also averts highly dynamic monomer fluctuations that could lead to fortuitous dimerisation[54]. The bulk of the amino acids involved in the head-to-head interaction are deleted in EGFRvIII[44], providing a rationale for the transformation potential of this mutant[59].

Conformational coupling was demonstrated in the ligand-bound state[40]. Here, in the ligand-free state, it is supported by the finding that kinase-mediated dimerisation and its requirement to form TM helix dimers, breaks head-to-head dimers and polymers inducing across the plasma membrane the formation of two dimer species, both distinct from the head-to-head dimer, the architectures of which were previously reported by long-timescale MD simulations[23] and EM experiments[26]. One kinase-mediated extracellular dimer is upright and extended, and consistent with a back-to-back conformation[23]. Results for L680N-EGFR together with MD simulations for the T766M mutant exploiting fully converged free-energy landscapes validate the prediction that the extended dimer is structurally coupled to the sTKD dimer[23]. The

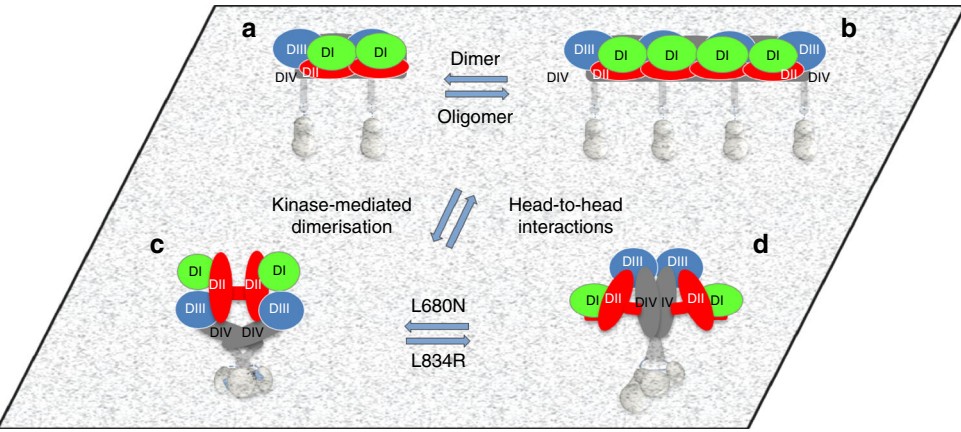

**Fig. 8** Cartoon models of ligand-free EGFR species on the cell surface. **a, b** Autoinhibited ligand-free receptors form **a** dimers and **b** larger oligomers via extracellular head-to-head interactions. Within head-to-head dimers and oligomers the ICMs remain as non-interacting units. **c, d** Kinase-mediated receptor dimerisation outcompetes head-to-head interactions to form two types of receptor dimers that typically coexist in equilibrium (bearing aTKD and sTKD dimer configurations). Head-to-head dimers and oligomers are disrupted by kinase-mediated dimerisation independently of whether the driver mutation and/or treatment is activating or not. The ECM architecture of one dimer type is consistent with a back-to-back dimer and structurally coupled to an sTKD dimer[6,22] (**c**). The ECM architecture of the other is consistent with a stalk-to-stalk dimer and structurally coupled via an N-terminal TM crossing to the aTKD dimer[40] (**d**). The L680N kinase domain mutation shifts the equilibrium toward the dimer population bearing sTKD dimers while L834R shifts the equilibrium towards the dimer population bearing the aTKD dimer. For all panels, DI is in green, DII in red, DIII in blue, DIV, TMD and JMD in grey, TKD in silver

second is a stalk-to-stalk dimer formed by DIV–DIV contacts and structurally coupled to the aTKD dimer[26]. Our results suggest that kinase-mediated dimer conformations coexist in equilibrium in constitutively active and in inactive receptor dimers, regulated by C-terminal interactions[54,56].

One key question remains. If the inactive sTKD dimer readily forms in constitutively active receptors, but not significantly in wtEGFR, what is its role if not autoinhibitory? It is tempting to speculate that the ligand-free EGFR structures we propose may be applicable to other HER family members. Indeed, recent results[60] show that, by stabilising a particular HER2 conformer, the TKI lapatinib drives the formation of a HER2–HER3 kinase domain heterodimer via a symmetric interface. The lapatinib-induced HER2–HER3 heterodimer is primed to respond to neuregulin, leading to signal amplification. It will be interesting to ascertain the possibility of an analogous role for the sTKD dimer.

## Methods

**Cell culture**. All reagents unless otherwise stated were from Invitrogen, UK. CHO cells expressing wtEGFR or the ΔC-EGFR, Δ973-EGFR, K721A-EGFR, Δ698-EGFR, L680N-EGFR mutants under an inducible Tet-ON promoter[35,61] were a gift from Prof. Linda Pike (Washington University, St. Louis). They were grown in 5% $CO_2$ in air at 37 °C in phenol-red free Dulbecco's modified Eagle's medium (DMEM) supplemented with 10% (v/v) foetal bovine serum, 2 mM glutamine, 100 µg/ml hygromycin B, and 100 µg/ml geneticin. Cells transiently transfected with I942E-EGFR, L834R-EGFR, IIIV/KKRE-EGFR, control wtEGFR or T766M-EGFR were allowed to grow for 48 h after transfection. All cells used were regularly tested for mycoplasma contamination.

**Drug treatments**. For all relevant experiments cells were treated with 1 µM erlotinib (Tocris Bioscience) in medium containing 0.1% serum during starvation and labelling, and imaging (for live cells). Cholesterol depletion was performed by incubating cells with 10 mM methyl beta-cyclodextrin (MβCD) (Sigma) in Opti-MEM (Invitrogen) for 30 min at 37 °C after the starvation step. Cells were rinsed twice with medium with 0.1% serum prior to labelling.

**Fluorophore localisation imaging with photobleaching**. The method was first described in ref. [28]. Briefly $1 \times 10^5$ CHO cells were seeded on 1% BSA-coated 35 mm no. 1.5 (high tolerance) glass-bottomed dishes (MatTek Corporation, USA) in 2 ml of media plus 50 ng/ml of doxycycline hyclate (Sigma), resulting in expression of ~$10^5$ receptors/cell[35]. After 48 h the medium was changed to 0.1% serum plus 50 ng/ml doxycycline for 2 h. Cells were rinsed and cooled to 4 °C for 10 min and then labelled with 4 nM CF640R-EGF or CF640R-Affibody, or with 8 nM EgB4-NB for 1 h on ice. The N-terminus of EGF was labelled at a 1:1 ratio by Cambridge

Research Biochemicals (Cleveland, UK). The EGFR Affibody was labelled at a 1:1 ratio at its single cysteine residue. EgB4-NB was conjugated to CF640R-NHS ester (Biotium) according to the manufacturer's instructions. Cells were rinsed and fixed with 3% paraformaldehyde (Electron Microscopy Sciences, USA) plus 0.5% glutaraldehyde (Sigma) for 15 min at 4 °C, then 15 min at room temperature. If required, cells were pre-treated for 1 h on ice at 4 °C with 200 nM 9G8-NB dissolved in PBS, or with erlotinib or MβCD as described above. We used an Axiovert 200M microscope with TIRF illuminator (Zeiss, UK), with a ×100 oil-immersion objective (α-Plan-Fluar, NA = 1.45; Zeiss, UK) and an EMCCD (iXon X3; Andor, UK). The microscope is also equipped with a wrap-around incubator (Pecon XL S1). Samples were illuminated with a fibre-coupled laser combiner (Andor) with a 100 mW 640 nm diode laser (Cube, Coherent). Images were collected every 0.28 s. Typically, for each experiment, approximately 120,000 single particle image spots were obtained from at least 750 cells and at least three biological repeats. Empirical posterior FLImP distributions were then obtained based on Affibody, EGF, or EgB4-NB separation measurements.

**FLImP decomposition and post-analysis**. The method has been described in detail in ref. [12]. Briefly, a FLImP histogram is a sum of multiple FLImP measurement empirical posterior distributions, and can be modelled as sum of Rician-distributions, one for each discrete underlying separation. We use a Bayesian parameter estimation to fit a given number of Rician components to the FLImP distribution, thus inferring the underlying distances, their uncertainties, and the number of measurements for each component. We use a BIC to choose objectively the number of components justified by the data. We incorporate bootstrap-resampling of the measurements to account for the effect of the finite number of measurements on the error bars and BIC test.

**Photo-bleaching imaging correlation spectroscopy**. Fluorescent images of the labelled receptor at the basolateral membrane were repetitively scanned with an Olympus FV1000 laser scanning confocal microscope to induce partial photo-bleaching. The resulting images were subjected to pbICS as described in ref. [38]. Briefly, for the "model-free" analysis, an aggregation model with monomer, dimer, trimer, tetramer, pentamer, hexamer, septamer, and octamer was assumed, where the fraction of each oligomeric species was variable and allowed to float in the fitting of the pbICS curves. The reported histogram is the cumulative average of seven pbICS curves per treatment condition, acquired over three independent biological replicates.

The derived cluster density (CD) as a function of fractional intensity remaining after each round of photobleaching ($p$) is given by Eq. (1).

$$CD(p) = \frac{\left(\sum_j j c_j\right)^2 p}{\sum_j \left[j c_j + j(j-1) c_j p\right]},\qquad (1)$$

Where $j$ is the oligomer size ($j = 1$ for monomer, $j = 2$ for dimer, etc.) and $c_j$ is the molar concentration of an oligomer size $j$. In the fitting procedure, we fixed $j=1, 2,$

3, 4, 5, 6, 7, 8 and allowed the concentrations $C_1...C_8$ to vary[62]. A typical raw dataset (CD versus $p$) together with fit is shown in Supplementary Fig. 3.

**FRET distance of closest approach.** Uncoated 35 mm no. 1.5 glass-bottomed dishes were seeded with $1 \times 10^5$ CHO cells expressing wtEGFR in 2 ml of media with 250 ng/ml of doxycycline (~$4 \times 10^5$ receptors/cell). After 48 h, the medium was changed to 0.1% serum with the same concentration of doxycycline for 2 h. Samples were labelled with 5 µM $C_{18}$ DiI (Thermo Fisher Scientific) for 10 min at 37 °C, then with either 100 nM EGFR Affibody or 200 nM EgB4-NB both labelled with Alexa Fluor 488 (custom conjugation) at 4 °C on ice for 1 h. Samples were fixed with 3% paraformaldehyde plus 0.5% glutaraldehyde as described above. The ensemble averaged DOCA of EGFR-bound Alexa 488 probes to cell membranes loaded with the fluorescent lipid analogue $C_{18}$DiI was estimated from the measured FRET efficiencies for increasing acceptor density within the membrane using the procedure described previously[32]. Briefly, bootstrap regression analysis was performed by randomly resampling datasets with replacement 3000 times and fitting each resampled dataset to the FRET response generated by Monte-Carlo simulation of a donor placed at increasing distances above a plane of acceptors. For each dataset a distribution of possible DOCAs is generated and the mean and standard deviation of these distributions are reported. Bootstrap hypothesis testing was used to compare differences between mean DOCAs. For each pair of datasets compared, bootstrap regression analysis was performed by resampling with replacement from the pool of the combined data points, two new datasets of equal size to the originals. The test statistic was calculated for each of 10,000 bootstrap replicates to generate a test distribution corresponding to the null hypothesis that the difference between the mean DOCA values is zero. The $p$-value reported is the proportion of the test distribution larger than the observed test statistic. The number of data points (each point from an individual cell) for wtEGFR + Affibody, wtEGFR + EgB4-NB, wtEGFR + 9G8-NB + Affibody, wtEGFR + 9G8-NB + EgB4-NB, wtEGFR + erlotinib + Affibody, wtEGFR + erlotinib + EgB4-NB, L680N-EGFR + Affibody, L680N-EGFR + EgB4-NB, L834R-EGFR + Affibody, and L834R-EGFR + EgB4-NB are 54, 63, 97, 30, 119, 170, 40, 47, 30, and 11, respectively. Data were taken from at least two independent biological replicates.

**Western blots.** CHO cells were seeded in six-well plates at a density of $1 \times 10^5$ cells/dish. 48 h later, cells were washed twice with ice-cold PBS chilled for 10 min then incubated on ice with ice-cold EGF for 1 h at 4 °C. Cells were washed twice with ice-cold PBS and scraped into PBS plus inhibitors (phosphatases and proteases) and spun down. Cells were lysed in 10× volume of M-PER lysis buffer (Pierce) + 100 mM NaF + 1 mM $Na_3VO_4$ + 1% protease inhibitors (Cell Signalling Technologies) + 150 mM NaCl + 1 mM EDTA at pH 8 and incubated for 10 min at room temperature. Cells were cleared by centrifugation and total protein measured. Sample buffer was added to 1× final concentration. Samples were run in parallel on 1.5 mm thick, 3–8% Tris-Acetate NuPAGE gels (Invitrogen) with HiMark Prestained HMW and Novex Sharp Prestained protein standards (Invitrogen) using an XCell apparatus (Invitrogen). Proteins were blotted using an iBlot system (Invitrogen) on PVDF membranes, blocked for 1 h at 4°C with 5% BSA in TBS + 0.1% Tween and probed overnight with mouse anti-EGFR pY1173 (Upstate (Millipore)—cat. no. 05–483) antibody diluted 1/1000. Gels were probed with secondary anti-mouse-HRP antibody (Jackson ImmunoResearch—cat. no. 715-035-150) diluted 1/1000 and incubated with Supersignal West Pico Chemiluminescent Substrate solution (Pierce) for 5 min, then imaged with a BioRad ChemiDoc MP system imager. Each blot was stripped with 25 ml stripping buffer (2% SDS, 0.75% β-mercaptoethanol, 62.5 mM Tris HCl pH 6.7) for 50 min at 60 °C, and re-probed with an anti-EGFR cocktail composed of anti-EGFR D38B1 (Cell Signalling Technologies—cat. no #4267), anti-EGFR N-Terminal polyclonal ab137660 (abcam), and anti-EGFR polyclonal 10005: sc-03 (Santa Cruz Biotechnology), each diluted 1/2000, all derived from rabbit. Anti-rabbit HRP (Jackson ImmunoResearch—cat. No 711-035-152) diluted 1/1000 was used for all blots and images were acquired as above. Densitometry was performed using Fiji[63] to extract the intensity of the bands and then calculating the ratio between the intensity of the pY1173 EGFR band and the corresponding total EGFR band for each condition. Experiments were performed in triplicate and data were presented in a box and whisker plot, showing the inclusive median, with the 25th and 75th quartile as box limits and the first and last quartile as whiskers.

**Tracking of EGFR complexes on CHO cells.** Cells were seeded at a density of $1 \times 10^5$ cells/dish on 1% BSA-coated 35 mm no. 1.5 (high tolerance) glass-bottomed dishes (MatTek Corporation, USA) in 2 ml of media. Receptor expression in CHO cells was induced with 50 ng/ml doxycycline. Prior to imaging, cells were starved for 2 h at 37 °C in 0.1% serum supplemented with 50 ng/ml doxycycline if required. Treatments with erlotinib and MβCD were performed as described above. Cells were then rinsed twice with 0.1% serum without doxycycline pre-heated at 37 °C and were labelled with a 1:1 mixture of 8 nM Alexa 488-Affibody/CF640R-Affibody for 10 min at 37 °C. Cells were rinsed twice with low serum medium without doxycycline pre-heated at 37 °C and promptly imaged. Single-molecule images were acquired using an Axiovert 200M microscope with a TIRF illuminator (Zeiss, UK), with a ×100 oil-immersion objective (α-Plan-Fluar, NA = 1.45; Zeiss, UK) and an EMCCD (iXon X3; Andor, UK). The microscope is also equipped with a wrap-around incubator (Pecon XL S1). The 488 and 642 nm lines of a LightHub

laser combiner (Omicron Laserage GmbH) were used to illuminate the sample and an Optosplit Image Splitter (Cairn Research) was used to separate the image into its spectral components as described previously[64]. The field of view of each channel for single-molecule imaging was $80 \times 30$ µm. Typically, for each condition, at least 30 field of views comprising one or more cells were acquired from a total of at least 3 independent biological replicates. All single-molecule time series data were analysed using the multidimensional analysis software described previously[31]. Briefly, this software performs frame-by-frame Bayesian segmentation to detect and measure features to sub-pixel precision, then links these features through time to create tracks using a simple proximity-based algorithm. The software determines cubic polynomial registration transformations from images of fluorescent beads. Feature detection and tracking was performed independently in each channel.

**Calculation of colocalisation and $\tau_{ON}$.** Two-colour TIRF images of the basolateral surfaces of cells were chromatically separated by a beam splitter and registered using custom made software to map the relative positions of the probes over the time course of data acquisition[31]. Single particle tracks were extracted as above[31]. The fraction of co-localised tracks was reported by tracks in which a receptor particle in one channel colocalises with another particle in the other channel, spending at least five 50 ms frames in total moving together within a pixel of each other. Coincidental colocalisation statistics were calculated for a dataset as follows. In each channel, a randomised set of tracks of the same size as the measured set was produced, where each track in the random sample was chosen (with replacement) from the measured tracks for that channel, recentred at random with uniform probability density across the field of view, rotated through a random angle with uniform probability density between 0 and 360°, and randomly flipped in $x$ with probability 0.5. The random tracks therefore have key properties such as durations and path structures representative of the true tracks, but now randomised in distribution and orientation. The colocalisation statistics were then calculated for the randomised tracks. This was performed a total of 50 times for each dataset, and the colocalisation statistics pooled to give a final estimate coincidental colocalisation fraction for that dataset. The reported colocalisation fraction is then the fraction for the real data minus the estimated coincident colocalisation fraction. The duration ($\tau_{ON}$) of individual events in which a track in one channel moves within a pixel of a track in the other channel and then they move apart again was also calculated. To reduce the impact of localisation error on these results a temporal Gaussian smoothing filter of FWHM 4 frames (200 ms) was applied to the position traces before the colocalisation analyses.

**9G8 nanobody binding affinity for different receptor conformations.** Ectodomain conformation was investigated by measuring the binding of 9G8-NB, which stabilises the tethered conformation, using the ratio between the binding of CF640R-9G8-NB, a probe which is selective for the tethered state of the receptor, and Alexa 488-EgB4-NB, a probe which is not, as determined by quantitative confocal microscopy. Briefly, uncoated 35 mm no. 1.5 glass-bottomed dishes were seeded with $1 \times 10^5$ CHO cells expressing wtEGFR in 2 ml of media with 250 ng/ml of doxycycline (~$4 \times 10^5$ receptors/cell). After 48 h, the medium was changed to 0.1% serum with the same concentration of doxycycline for 2 h and cells were treated with 1 µM erlotinib as described in the drug treatments section above, if required. Cells were then treated with 50 nM Alexa 488-EgB4-NB for 1 h at 4 °C and then fixed with 3% PFA for 15 min at 4 °C, followed by 15 min fixation at room temperature. This was done to prevent 9G8-NB from inducing conformational changes. After fixation, cells were washed three times with PBS and then treated for 1 h at 4 °C with 200 nM of CF640R-9G8-NB. Twenty confocal images of equatorial regions of the cells were collected from three replicates, using a Leica TCS SP8 microscope with a (NA = 1.4; Leica) and a Leica HyD hybrid detector. Fluorescence was excited with 488 or 640 nm light produced by an NKT Extreme supercontinuum light source. Using FIJI, regions of interest corresponding to the plasma membrane of individual cells were drawn manually. The mean Alexa 488 and CF640R fluorescence were measured and the ratio of CF640R:Alexa 488 fluorescence intensity calculated for each cell.

**MD simulations of the head-to-head EGFR dimer on membrane.** Simulations were run for two set-ups: with and without VHH nanobody. Each EGFR monomer in the simulations contained residues 3–650; the VHH nanobody in the model contained residues 1–124. The ECM of EGFR monomer was obtained from PDB entry 1NQL[5], and loops were built manually to connect the extracellular domains and transmembrane helix. The EGFR monomers were assembled into a dimer based on crystal packing contact observed in PDB entry 4KRP[42]. A VHH nanobody was added to each EGFR monomer based on the EGFR–VHH complex in PDB entry 4KRP in the simulation of VHH-bound EGFR dimer. The protein complexes were placed in/on a membrane consisting of only 1-palmitoyl-2-oleoylphosphatidylcholine lipids. The above systems were solvated in water with 0.15 M NaCl. Additional $Cl^-$ ions were included to neutralise the net charges of the proteins (+2 total). The simulation systems were parameterised using the Amber ff99SBstar-ILDN force field for proteins[65], the CHARMM 36 force field for lipids and ions[66], and TIP3P for water[67]. After relaxation, the system was further equilibrated in the NPT ensemble for 1 ns using GPU Desmond. The simulation system of EGFR dimer without VHH nanobody was $17.9 \times 17.9 \times 17.2$ nm$^3$ in dimensions

and contained 550,717 atoms. The simulated system of EGFR dimer with VHH nanobody was $21.5 \times 21.5 \times 19.8$ nm$^3$ in dimensions and contained 879,010 atoms. The simulations were then performed on a special-purpose supercomputer, Anton 2 (ref. [68]). Simulations of the EGFR dimer without VHH nanobody ran for 20.0 and 13.8 μs in two repeats, and simulations of the EGFR dimer with VHH nanobody ran for 20.0 μs in two repeats. Production simulations were performed in the constant number (N), pressure (P, semi-isotropic), and temperature (T) (NPT) ensemble with $T = 310$ K and $P = 1$ bar using a variant of the Nosé–Hoover[69] and the Martyna–Tobias–Klein algorithm[70]. The RESPA integration method was used with time step of 2.5 fs. Long-range electrostatics evaluated every three time steps using the u-series method with a 1.1–1.3 nm cutoff for the electrostatic pairwise summation: a 0.9 nm cutoff for the van der Waals calculations. Water molecules and all bond lengths to hydrogen atoms were constrained using M-SHAKE[71].

**MD simulations and free-energy calculations**. The X-ray structures used as models for the wild type and T766M mutant sTKD dimer are deposited under the Protein Data Bank (PDB) codes 5CNO (ref. [6]) and 4I24 (ref. [72]), respectively. The missing atoms were built with MODELLER[73]. Both dimers consist of the sequence N700-D1014 (in the numbering with the 24-aa tag). For unbiased MD simulations, each simulated system was generated using AMBER14SB force field[74] at pH 7.4. The protonation states of the residues were determined by PROPKA3.0 (ref. [75]), which left all the residues in their usual charge states, except for His805 which was protonated. The systems were solvated with ~36,000 TIP3P water molecules[76] in a dodecahedral box with periodic boundary conditions, while Na$^+$ and Cl$^-$ ions were added to reach neutrality and the final concentration of 0.15 M (the total number of atoms was ~120,000). The production simulations were generated using GROMACS 5.1.4 biomolecular simulation package[77] with a 2-fs integration step, constant temperature of 310 K using velocity rescale thermostat, and constant pressure of 1 bar. Further simulation details can be found in Supplementary Methods. For free-energy calculations, fully converged free-energy landscapes of the wild type and single mutant monomers were obtained by parallel tempering metadynamics[78] simulations in the well-tempered ensemble (with 6 replicas per system and ~680 and ~610 ns per replica, respectively)[34]. The free energies were initially projected as a function of three collective variables: the difference between two salt-bridge distances: d(K721, E738)–d(K721, D834) which were found to switch during the activation, and two contact maps (one corresponding to the Src-like inactive state and the other corresponding to the active extended state). FESs as a function of a variable that captures the α-helical content of the protein (ALPHARMSD with default settings) and a contact map corresponding to the active extended A-loop conformation (distance from the reference structure) were obtained by reweighting the trajectories and the converged FESs[34] using the PLUMED 2.3.2 library[79]. Further analysis details can be found in Supplementary Methods.

**Plasmids and transfections**. The I942E-EGFR pcDNA3 plasmid and IIIV/KKRE-EGFR plasmid were gifts from Prof. J. Kuriyan (University of California Berkley). The wtEGFR pCDNA3 plasmid used as a control in the western blot experiments was a gift from Prof. Y. Yarden (Weizmann Institute of Science). The L834R-EGFR pCDNA3 and T766M-EGFR pCDNA3 plasmids were produced through site-directed mutagenesis using the primers described in Supplementary Table 5. Reactions were performed using a site-directed mutagenesis kit from Stratagene following the manufacturer's instructions. EGFR IIIV/KKRE was amplified by PCR using primers containing infusion tags (described in Supplementary Table 5) for the insertion of the PCR product into the pOPINE vector[80] cut with NcoI and PmeI (gift from Prof Ray Owens, University of Oxford). The In-Fusion® HD EcoDry™ Cloning Plus kit (Takara) was used according to the manufacturer's instructions. The final product was confirmed by complete sequencing. Transfections were performed in solution at the same time as seeding, using Viafect (Promega) as a carrier for the I942E-EGFR, L834R-EGFR, T766M-EGFR and wtEGFR plasmids. GeneJuice (Merck Millipore) was used as a carrier for IIIV/KKRE-EGFR. In all cases expression was allowed to proceed for 48 h prior to further experimental manipulations.

## Data availability

Data supporting the findings of this manuscript are available from the corresponding author on reasonable request.

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

## Acknowledgements

We thank Prof. John Kuriyan for the generous gift of the I942E and the IIIV/KKRE mutant constructs; Prof. Linda Pike for the generous gift of the CHO cells maintained by an inducible Tet-ON promoter expressing the various EGFR mutants; and Prof Yosef Yarden for the gift of the pCDNA3 WT EGFR plasmid. Mr. Teodor Boyadzhiev and Dr. Michael Hirsch for technical support. This work has been funded by MRC grant (Ref. MR/K015591/1) from the Medical Research Council and by BBSRC grant BB/G006911/1 from the Biotechnology and Biological Sciences Research Council. F.L.G. and A.K .acknowledge EPSRC for financial support (grants EP/M013898/1; EP/P022138/1; EP/P011306/1).

## Author contributions

L.C.Z.-D., D.K., S.R.N., P.J.P., and M.L.M.-F. have devised the study and interpreted the data. L.C.Z.-D. has performed the SPT experiments. L.C.Z.-D. and E.O.-Z. performed and analysed western blots; D.K. has performed and analysed the 9G8-NB binding experiments; S.R.N., D.K., S.K.R., and L.C.Z.-D. have performed the FLImP experiments and analysed the data; C.J.T. has performed and analysed the DOCA experiments; S.S., Y. S., and D.E.S. carried out the molecular dynamic simulations of the head-to-head dimer. A.K. and F.L.G. carried out the molecular dynamic simulations of the sTKD dimer and free-energy landscape calculations of the monomer. S.K.R. has re-cloned the IIIV/KKRE

plasmid into a mammalian expression vector; P.J., R.C.R., and P.vB.eH. have produced and tested the EgB4-NB and 9G8-NB and carried out the tests to show specific binding of EgB4-NB to DI; A.L. and A.H.A.C. have performed the pbICS experiments and analysed the data; D.T.C. has performed the Affibody, EgB4-NB, and 9G8-NB conjugations; D.J.R. has written, tested, and implemented the automated data analysis routines used to analyse the SPT and FLImP data and performed the SPT analysis and FLImP post-analysis; E.O.-Z. and G.S. have produce the L834R and T766M mutants and carried out functional testing on them. P.J.P. has contributed to data interpretation and to discussions about the models of EGFR complex formation. M.L.M.-F., D.T.C., and P.J.P. have scoped and written the paper.

## Additional information

**Competing interests:** The authors declare no competing interests.

