## [Peer Review File · Nature Communications]

Reviewers' comments:

Reviewer #1 (Remarks to the Author):

The epidermal growth factor receptor (EGFR) is a prototypical cell surface growth factor receptor that plays critical roles in both normal cellular growth and development and abnormal cancerous growth. While EGFR regulation has been extensively studied, the mechanisms by which its kinase activity is suppressed in the basal state (i.e. in the absence of ligand and oncogenic kinase-activating mutations) are not fully understood. This manuscript presents an extensive characterization of the oligomerization of the EGFR occurring in the basal state, which the authors conclude plays a critical role in the autoinhibition of the EGFR kinase activity. The study employs a diversity of experimental approaches that together are well suited to interrogate the oligomeric interactions of the receptor, with associated experimentation addressing the structural basis of oligomerization. Ultimately, the authors propose a model for oligomerization in the basal state that invokes extracellular domain intracellular domain (ECD) interactions only and actually precludes intracellular domain (ICD) interactions, thus ensuring kinase inhibition in the basal state. This model could constitute a significant advancement in our understanding of EGFR regulation.

The data presented appear to be of high quality. An exception might be the FRET assays of distance to the membrane of different subdomains of the ECD, in which case, no doubt for technical reasons, the data points have rather large error bars and do not seem to conform well to the simulated curves used in their analysis (Supplemental Figure 5). Perhaps the authors can explain how these experimental data yielded distance determinations of such high apparent precision (Figures 4E and 5H).

Other significant concerns of this reviewer include some interpretations of specific results, and some of the language use in their discussion. Specific comments and queries related to this include the following.

The authors present three molecular simulations of the ECD interactions in the head-to-head dimer including that of Figure 3C, in which much of subdomain I was deleted from the model. However, they say nothing about the purpose of this particular simulation and what the results of it showed. Also, in discussing these same simulations (lines 148 and 161), the authors twice refer to the development of a "trans interaction between DI and DII." However, especially in the case of Figure 3D, it does not appear that there is an intermolecular interaction between domains DI (green) and DII (red) of the different ECD monomers and in fact these two domains appear to be widely separated in space.

In interpretation of the various FLImP analysis the authors frequently refer to the 5 nm and 13 nm components in DIII-DIII separation histograms as reflective of "stalk-to-stalk dimers" and "extended dimers" (cf. Figure 6 legend), which have specifically defined structures (Figure 1E and 1D, respectively) distinct from that of the proposed head-to-head dimer. However, the smaller-sized components of separation seen in the case of the wild-type EGFR (5 nm and 13 nm) and the EGFR-deltaC mutant (5-6 nm and 13-15 nm) (Figures 6C and 4C) are the same or not very different from the distances of separation of monomers in the head-to-head oligomers predicted by the model of Figure 4A (7-8 nm and 14-15 nm). Indeed these smaller components in the case of Figure 4C are referred to as representing monomer separations in the head-to-head oligomer (cf. lines 179-180). In their discussion of Figure 6D (lines 298-301), the authors seem to imply that the 5 nm component in the case of the L858R-EGFR represents the aTKD dimer. Regarding Figure 6E (lines 305-307), the authors state "The 5 nm and 13 nm components were commensurate in T790M-EGFR and delta973-EGFR, whose trans-phosphorylation depends upon the formation of the aTKD." It is unclear what is the meaning of "commensurate" here. However, it seems that the authors can "see" different underlying structures in the same-sized components of the FLImP histograms obtained with the different EGFR receptor constructs.

In regard to identification of the 13 nm component as the "extended dimer" species, the authors should indicate whether this distance is or is not consistent with that predicted from a structural model, such as those used in supplementary Figure 3 for evaluating distances of separation in the head-to-head dimer model or the oligomer model as presented in Figure 4A.

Minor concerns:

In the case of the histogram of Figure 4D the authors state that the "main components (were) resolved consistent with model predictions" (line 184). However, one component (albeit not major) is resolved at 38 nm, which is larger than the predicted by the model (Figure 4A).

In the phrase of lines 246-247, "In conclusion, the close DIV-DIV contacts revealed by Western Blot...", it is entirely unclear to what experiments the authors are referring.

In supplemental Figure 3, it appears that panels A and B are misidentified in the legend.

Reviewer #2 (Remarks to the Author):

The article by Zanetti-Domingues is an extensive study of different EGFR oligomers by biochemical, microscopy and computational methods. Using a range of mutants to differentiate between C-terminus, TKD and ECM influences on dimerization, the authors build a model of different EGFR oligomer conformations that bring together the results of various reported experiments in the past and that help elucidate why various research labs have seen pre-formed dimers. This is a well written article that advances our understanding of the mechanism of action of EGFR.

Comments/Questions

In the FLImP measurements, is any filtering performed on the localized spots? If yes that should be described as in the original articles the authors indicated that spots with large confidence intervals are removed to obtain better separation statistics.

What is the 5th component in Fig. 4c? It is detectable but is not explained by the authors.

In Fig. 4 there are more components for DI-DI separations than for DIII-DIII. Why is that the case?

If in wtEGFR DI and DIII have similar distances from the membrane but DI-DI and DIII-DIII distances are different. In that case, would the curvature of the oligomer not lie in the membrane plane? But that would not provide an advantage for incorporation into vesicles as the authors claim.

In regard to pre-formed dimers, I think it would be fair to cite the work of Maruyama who has shown large scale dimerization early on (Moriki et al, JMB, 2001) and whose work has shown very similar dimerization levels (~67% of all monomers in dimers: Liu et al, Biophys. J., 2007)

Minor issues:

In line 190, DiIC18 is described as DID? I assume that is a spelling mistake.

In line 221, replace "actives" by "activates"

Reviewer #3 (Remarks to the Author):

I have read "The architecture of the EGFR's basal complexes reveals autoinhibition mechanisms in dimers and oligomers" by Zanetti-Domingues et al. In this work, the authors make a Herculean effort aimed at recovering distances between EGFR domains and the plasma membrane as well as the oligomer status of the EGFR to generate a structural model of the EGFR and its complexes at the cell surface. The number of measurements made are remarkable and the data presented along provide insight into the structure of the EGFR. Only reviewers that are EGFR aficionado's would have the depth of knowledge to evaluate all of the different mutants and conditions used in this study. That being said, the data are overall (and particularly the FLImP data are consistent and seem to indicate structural rearrangements that the authors interpret by drawing on significant background.

This reviewer has a few significant concerns regarding the experiments and their interpretation:

Major point – how does expression level change the behavior of the EGFR dimerization and oligomerization? In particular, photobleaching ICS was used to determine the number of monomer and dimers formed and some discussion was provided regarding how increasing the number of EGFR changes the multimerization on the cell surface, and ultimately drives spontaneous phosphorylation of the receptor. Some measurement of how this influences the results would be very helpful for making the case that this is a generalizable observation.

Conclusions are drawn regarding the movements of the domains of the EGFR relative to the plasma membrane using the FRET methods of Tynan et al. Mol. Biol. Cell 2011. This approach requires fitting of data over various acceptor (DiD) concentrations to obtain an average distance between the label fluorophore and the plasma membrane. The data in supplemental figure 5 that generate the bar graphs in Fig 4 and 5, is not very confidence inspiring. In fact, it appears that if the regressions were not anchored at – many would have a negative slope, suggesting that the model of Tyrnan et. al may not apply. Furthermore, the low FRET efficiencies and limited acceptor density call into question the validity and interpretation of these results. Can additional experiments be completed that would shore up the FRET results (such as trying to obtain a higher acceptor density)? Are these experiments essential for making the structural argument?

Minor point:

The panel in Figure 2b is labeled DI-DI, whereas legend and results indicate this is the DIII-DIII separation distance

It is unclear to this reviewer how photobleaching ICS gave estimates the monomer fraction. It would be nice to have the raw data as a supplement. E.g. more data on the correlation function and regression for this measurement would be helpful.

Reply to reviewers, manuscript ref NCOMMS-18-14237: 'The architecture of EGFR's basal complexes reveals autoinhibition mechanisms in dimers and oligomers'

NOTES TO REVIEWERS: We thank the reviewers for their insightful comments. We have revised the manuscript accordingly and believe it is now clearer and more robust. Changes in the revised manuscript are shown in tracking mode and described below under the comments. Because the version submitted last May already had the maximum word count (5,000), we have tidied the English in the manuscript to accommodate the additional explanations required. New figures have been added in the main text and supplementary file to address the points raised. Figure numbers below refer to the figures as they appear in this revised version. Please note that highlighted line numbers are as they appear in the PDF file that contains all changes tracked ["Tracked manuscript with line numbering used in reply to reviewers.pdf"]. A word document with all changes accepted was also submitted. Lastly, because it is now common to number mutations without the 24 amino acid signalling peptide, we have changed accordingly the numbering of mutants in this revised version.

Reviewer #1 (Remarks to the Author):

The epidermal growth factor receptor (EGFR) is a prototypical cell surface growth factor receptor that plays critical roles in both normal cellular growth and development and abnormal cancerous growth. While EGFR regulation has been extensively studied, the mechanisms by which its kinase activity is suppressed in the basal state (i.e. in the absence of ligand and oncogenic kinase-activating mutations) are not fully understood. This manuscript presents an extensive characterization of the oligomerization of the EGFR occurring in the basal state, which the authors conclude plays a critical role in the autoinhibition of the EGFR kinase activity. The study employs a diversity of experimental approaches that together are well suited to interrogate the oligomeric interactions of the receptor, with associated experimentation addressing the structural basis of oligomerization. Ultimately, the authors propose a model for oligomerization in the basal state that invokes extracellular domain-intracellular domain (ECD) interactions only and actually precludes intracellular domain (ICD) interactions, thus ensuring kinase inhibition in the basal state. This model could constitute a significant advancement in our understanding of EGFR regulation.

The data presented appear to be of high quality. An exception might be the FRET assays of distance to the membrane of different subdomains of the ECD, in which case, no doubt for technical reasons, the data points have rather large error bars and do not seem to conform well to the simulated curves used in their analysis (Supplemental Figure 5). Perhaps the authors can explain how these experimental data yielded distance determinations of such high apparent precision (Figures 4E and 5H).

We agree with the reviewer: some FRET assays were below par and the reasons are those mentioned. We have carefully repeated the relevant experiments and managed to get better statistics. We have also explained how the precision of the measurements is calculated. In addition, we added data to the main text validating FRET results by using confocal microscopy to measure the degree of 9G8-NB binding to probe conformation. New results are: (i) in the Supplementary file - Supplementary Fig. 7 and Supplementary Table 1; (ii) in main text - Fig. 4e, Fig. 5j, 5k.

Other significant concerns of this reviewer include some interpretations of specific results, and some of the language use in their discussion. Specific comments and queries related to this include the following.

The authors present three molecular simulations of the ECD interactions in the head-to-head dimer

including that of Figure 3C, in which much of subdomain I was deleted from the model. However, they say nothing about the purpose of this particular simulation and what the results of it showed.

We can easily clarify this. Fig. 3c is not a different simulation. It is the same simulation shown in Fig. 3b, in which we have manually deleted amino acids 6-273 see lines 160-161. The purpose of showing this in Fig. 3c is to highlight the important observation that the head-to-head interaction involves almost the same amino acids that are deleted in the constitutively active EGFRvIII, which causes glioblastoma. This observation suggests the head-to-head interface is biologically significant.

Also, in discussing these same simulations (lines 148 and 161), the authors twice refer to the development of a “trans interaction between DI and DII.” However, especially in the case of Figure 3D, it does not appear that there is an intermolecular interaction between domains DI (green) and DII (red) of the different ECD monomers and in fact these two domains appear to be widely separated in space.

We have added new figures showing these contacts (Fig. 3b-3d, right panels).

In interpretation of the various FLImP analysis the authors frequently refer to the 5 nm and 13 nm components in DIII-DIII separation histograms as reflective of “stalk-to-stalk dimers” and “extended dimers” (cf. Figure 6 legend), which have specifically defined structures (Figure 1E and 1D, respectively) distinct from that of the proposed head-to-head dimer. However, the smaller-sized components of separation seen in the case of the wild-type EGFR (5 nm and 13 nm) and the EGFR-deltaC mutant (5-6 nm and 13-15 nm) (Figures 6C and 4C) are the same or not very different from the distances of separation of monomers in the head-to-head oligomers predicted by the model of Figure 4A (7-8 nm and 14-15 nm). Indeed these smaller components in the case of Figure 4C are referred to as representing monomer separations in the head-to-head oligomer (cf. lines 179-180).

The reviewer is right to point out that the ~5 nm resolution of FLImP is not good enough to distinguish between different dimer structures (we indeed already acknowledged this in the previous version). This is unsurprising because at 5 nm resolution the structural details of different dimers of a molecule of the size of the EGFR cannot be ascertained. We suspect that the extracellular portions of kinase-mediated dimers were different because it is difficult to see how the N-crossing TM dimer could be accommodated by the head-to-head interface. We however agree that this was not very well explained.

Head-to-head dimers, stalk-to-stalk dimers and back-to-back dimers were distinguished as follows:

- **The three dimer structures have significantly different conformations reported by their distinct FRET signatures (Fig. 4e, 5j), Supplementary Fig 7a-7d, and Supplementary Table I (now with new data and error analysis added).**
- **The stalk-to-stalk and head-to-head dimer structures show different conformational flexibility, reported by their different degree of binding to a conformational probe (9G8-NB) (Fig. 5k)**
- **The three dimer structures show different dissociation kinetics (reported by SPT) (Fig. 5e, 5f)**

The results are eminently consistent with kinase-mediated dimerisation promoting the previously reported stalk-to-stalk and back-to-back dimers (Fig. 1d, 1e). To address this comment, we have extensively revised an entire results section in the manuscript, see lines 240-321.

In their discussion of Figure 6D (lines 298-301), the authors seem to imply that the 5 nm component in the case of the L858R-EGFR represents the aTKD dimer. Regarding Figure 6E (lines 305-307), the authors state “The 5 nm and 13 nm components were commensurate in T790M-EGFR and delta973-EGFR, whose trans-

phosphorylation depends upon the formation of the aTKD.” It is unclear what is the meaning of “commensurate” here. (We meant of similar sizes. We have changed the sentence to increase clarity, lines 380-381) However, it seems that the authors can “see” different underlying structures in the same-sized components of the FLImP histograms obtained with the different EGFR receptor constructs.

Our reasoning is as follows: When we classified the FLImP results according to the shape of the distributions (Fig. 6c-6d), L834R-EGFR (before L858R) displayed a dominant 5 nm peak and fewer oligomers. This is similar to wtEGFR + erlotinib and wtEGFR +MβCD, and significantly different to the two other groups. We know that the L834R mutation promotes the aTKD dimer (Shan et al, 2012). Our previous results (Fig. 5) predicted that aTKD formation would return a dominant DIII-DIII separation 5 nm component in L834R-EGFR, as indeed we found. In the case of wtEGFR + erlotinib and wtEGFR +MβCD we showed this 5 nm peak to report a stalk-to-stalk dimer (Fig. 5). We carried out new FRET experiments in L834R-EGFR which indicate the extracellular conformation of L834R-EGFR is different to that of wtEGFR and L680N-EGFR, but indistinguishable to that of erlotinib-bound wtEGFR (See Supplementary Fig. 7e, and Supplementary Table 1). In further support of this, the STP results for L834R-EGFR are consistent with those of wtEGFR in MβCD-treated cells (Fig. 7f). We therefore concluded that the dominant 5 nm peak in L834R-EGFR also represents stalk-to-stalk dimers. We have clarified this point in lines 359-374. We also like to point out that the notion of conformational coupling, which the results for L834R-EGFR support, was further validated in Fig. 6f by the MD simulations in T766M-EGFR (previously T790-EGFR)

In regard to identification of the 13 nm component as the “extended dimer” species, the authors should indicate whether this distance is or is not consistent with that predicted from a structural model, such as those used in supplementary Figure 3 for evaluating distances of separation in the head-to-head dimer model or the oligomer model as presented in Figure 4A.

There is no crystal structure of the Affibody bound to EGFR. We however know the separations between the EGF binding sites in the ligand-free extended dimer model are 7 nm (Arkhipov et al, 2013). The largest dimension of the Affibody is 3.7 nm. Therefore the 13 nm separation measured in the extended dimer is within range of $(2 \times 3.7 + 7)$ nm (lines 313-317)

Minor concerns:

In the case of the histogram of Figure 4D the authors state that the “main components (were) resolved consistent with model predictions” (line 184). However, one component (albeit not major) is resolved at 38 nm, which is larger than the predicted by the model (Figure 4A).

Under the small 38 nm component there are two separation measurements. FLImP, being stochastic, can report stoichiometric separations and ad hoc separations, the latter from receptors randomly near each other. The advantage of FLImP is that it is quantitative regarding both the separation components and their prevalence. We now show in Supplementary Fig. 6c-6g that the small 38 nm component becomes less significant as more data are retained. We also show that the differences between Fig. 4D and Fig. 4C are statistically significant (Supplementary Fig. 6h-6j).

In the phrase of lines 246-247, “In conclusion, the close DIV-DIV contacts revealed by Western Blot...,” it is entirely unclear to what experiments the authors are referring.

We referred at the Western Blot experiments that show that MβCD does not activate a mutant that cannot form DIV-DIV contacts (Fig. 5c). We have clarified this point in the manuscript (line 268-270).

In supplemental Figure 3, it appears that panels A and B are misidentified in the legend.

This has been corrected.

Reviewer #2 (Remarks to the Author):

The article by Zanetti-Domingues is an extensive study of different EGFR oligomers by biochemical, microscopy and computational methods. Using a range of mutants to differentiate between C-terminus, TKD and ECM influences on dimerization, the authors build a model of different EGFR oligomer conformations that bring together the results of various reported experiments in the past and that help elucidate why various research labs have seen pre-formed dimers. This is a well written article that advances our understanding of the mechanism of action of EGFR.

Comments/Questions

In the FLImP measurements, is any filtering performed on the localized spots? If yes that should be described as in the original articles the authors indicated that spots with large confidence intervals are removed to obtain better separation statistics.

As the reviewer points out, separations with smaller confidence intervals are retained on the basis of signal-to-noise to achieve a better resolution. However, in the absence of a bias, this does not affect results, as previously described in Needham et al, 2013, 2015, 2016; Webb et al, 2015; Zanetti-Domingues et al, 2015. We have included a sentence reiterating this and the appropriate references in the caption of Fig. 2a (lines 752-753).

What is the 5th component in Fig. 4c? It is detectable but is not explained by the authors.

As the reviewer points out, there is one separation at 56.3 nm (Supplementary Fig. 6a). FLImP, being stochastic, can report both stoichiometric separations and ad hoc separations, the latter from receptors randomly near each other. For this reasons, we have to be careful not to over-interpret results coming from single separations. We believe the separation at 56.3 nm is a random event because of the following reasoning: Given that the 4th component of the 6 nm resolution FLImP distribution in Fig. 4c (42 nm) is separated by 14 nm from the isolated separation at 56 nm, if the latter reported stoichiometric interactions, when 2-fold more data are retained at 7 nm resolution, a resolution which should still resolve components at 42 nm and 56 nm, we would still resolved the two components. The absence of any evidence of a peak component at 56 nm in data at 7 nm resolution (Supplementary Fig. 6b) suggests that the isolated 56.3 nm separation is not stoichiometric.

In Fig. 4 there are more components for DI-DI separations than for DIII-DIII. Why is that the case?

We are unsure what the reviewer means in this point. In Fig. 4c there are four DIII-DIII components and three DI-DI components in Fig. 4d. We can only speculate that the reviewer has meant to ask the reverse question, i.e. 'why there are more DIII-DIII components than DI-DI ones?' If this is the case, the answer is because the limited resolution of FLImP resolves fewer DI-DI separations, which are smaller and closer together, than DIII-DIII ones. This is the likely reason why the 13.3 DI-DI peak component in Fig. 4d is broader than typical DIII-DIII peaks (lines 192-195).

If in wtEGFR DI and DIII have similar distances from the membrane but DI-DI and DIII-DIII distances are

different. In that case, would the curvature of the oligomer not lie in the membrane plane? But that would not provide an advantage for incorporation into vesicles as the authors claim.

The reviewer is absolutely correct that the curvature lies in the plane of the membrane. We wrote: ‘would facilitate receptor packing ... at the mouths of caveolae’. By mouths of caveolae we meant around the entrance and not within. We have clarified this in lines 175-178.

In regard to pre-formed dimers, I think it would be fair to cite the work of Maruyama who has shown large scale dimerization early on (Moriki et al, JMB, 2001) and whose work has shown very similar dimerization levels (~67% of all monomers in dimers: Liu et al, Biophys. J., 2007)

This is done (refs 20 and 21) in the Introduction (line 64) and Results (line 113).

Minor issues:

In line 190, DilC18 is described as DID? I assume that is a spelling mistake. **(Corrected)**

In line 221, replace “actives” by “activates” **(Corrected)**

Reviewer #3 (Remarks to the Author):

I have read “The architecture of the EGFR’s basal complexes reveals autoinhibition mechanisms in dimers and oligomers” by Zanetti-Domingues et al. In this work, the authors make a Herculean effort aimed at recovering distances between EGFR domains and the plasma membrane as well as the oligomer status of the EGFR to generate a structural model of the EGFR and its complexes at the cell surface. The number of measurements made are remarkable and the data presented along provide insight into the structure of the EGFR. Only reviewers that are EGFR aficionados would have the depth of knowledge to evaluate all of the different mutants and conditions used in this study. That being said, the data are overall (and particularly the FLImP data are consistent and seem to indicate structural rearrangements that the authors interpret by drawing on significant background.

This reviewer has a few significant concerns regarding the experiments and their interpretation:

Major point – how does expression level change the behavior of the EGFR dimerization and oligomerization? In particular, photobleaching ICS was used to determine the number of monomer and dimers formed and some discussion was provided regarding how increasing the number of EGFR changes the multimerization on the cell surface, and ultimately drives spontaneous phosphorylation of the receptor. Some measurement of how this influences the results would be very helpful for making the case that this is a generalizable observation.

This is an important observation. In a previous publication we showed that in T47D cells, which express <10,000 EGFR copies/cell, the FLImP distribution of WT-EGFR bound to the same Affibody is indistinguishable to that found here in CHO cells expressing 10-fold more EGFR copies/cell (see Fig. 10G in Needham et al PLOS ONE 2013 doi:10.1371/journal.pone.0062331). This suggests that the shape of the oligomers formed by wtEGFR is independent of receptor numbers on the cell surface. We suggest that the result in T470D cells, with 10-fold fewer receptor numbers, is more relevant to answer the question asked by the reviewer because at ~10-fold higher number, EGFR becomes de-regulated, thus the complexes formed are not representative of the ligand-free inactive conditions we have reported in this manuscript. We have commented on this point on lines 105-107.

Conclusions are drawn regarding the movements of the domains of the EGFR relative to the plasma

membrane using the FRET methods of Tynan et al. Mol. Biol. Cell 2011. This approach requires fitting of data over various acceptor (DiD) concentrations to obtain an average distance between the label fluorophore and the plasma membrane. The data in supplemental figure 5 that generate the bar graphs in Fig 4 and 5, is not very confidence inspiring. In fact, it appears that if the regressions were not anchored at – many would have a negative slope, suggesting that the model of Tyrnan et. al may not apply. Furthermore, the low FRET efficiencies and limited acceptor density call into question the validity and interpretation of these results. Can additional experiments be completed that would shore up the FRET results (such as trying to obtain a higher acceptor density)? Are these experiments essential for making the structural argument?

We agree with the reviewer that some FRET assays were below par. We have carefully repeated the relevant experiments and are happy to report that we managed to get better statistics. The new results are: (i) in the Supplementary file - Supplementary Fig. 7 and Supplementary Table 1; (ii) in the main text - Fig. 4e, Fig. 5j, 5k. Associated text changes are in lines 274-279 and lines 310-321. The FRET results are important to distinguish between different types of dimers.

Minor point:

The panel in Figure 2b is labeled DI-DI, whereas legend and results indicate this is the DIII-DIII separation distance (**Corrected**)

It is unclear to this reviewer how photobleaching ICS gave estimates the monomer fraction. It would be nice to have the raw data as a supplement. E.g. more data on the correlation function and regression for this measurement would be helpful.

We have added in supplementary info a typical image and a typical curve with fit (Supplementary Fig. 2a-2b). We have also added in the methods section the formula we used to relate the pbICS to the aggregate distribution (which includes monomers) and reference the paper where that formula appears (lines 577-583).

REVIEWERS' COMMENTS:

Reviewer #1 (Remarks to the Author):

In reviewing the original manuscript, a main concern of this reviewer was the apparent poor agreement of the FRET distance of closest approach (DOCA) data with the simulated theoretical curves. While the new FRET data in the revised manuscript, do not entirely resolve this issue, it does seem that the raw data indicate significant changes in the FRET efficiency that are likely reflective of changes in the DOCA.

Other concerns regarded interpretation of results and wording in the text, which are satisfactorily addressed in the revised manuscript.

Reviewer #2 (Remarks to the Author):

The authors have answered my questions. I have no further comments.

Reviewer #3 (Remarks to the Author):

While the authors responded that the question "how does EGFR receptor density affect the multimerization and phosphorylation?" raised by rev 3, is important, the revised text does little to address this concern. The fact that between two different cell lines known to have different levels of EGFR expression does not immediately lead this reviewer to the conclusion that dimerization is 'independent' of EGFR concentration. Given that so many of the methods used here depend critically on this question, it would have been reassuring if a study using shRNA, inducible or disabled promoters were used to firmly demonstrate this case.

While the DiD-FRET measurements seem to be slightly improved, they still span a very narrow range in acceptor density and the statistical model does not seem valid. Shotgun blast data seems to be giving distance measurements with +/- 0.7 nm accuracy?

Reply to reviewers manuscript ref NCOMMS-18-14237: 'The architecture of EGFR's basal complexes reveals autoinhibition mechanisms in dimers and oligomers'

Reviewer #1 (Remarks to the Author): In reviewing the original manuscript, a main concern of this reviewer was the apparent poor agreement of the FRET distance of closest approach (DOCA) data with the simulated theoretical curves. While the new FRET data in the revised manuscript, do not entirely resolve this issue, it does seem that the raw data indicate significant changes in the FRET efficiency that are likely reflective of changes in the DOCA.

The noisy appearance of the FRET data reflects the intrinsic (and rather typical) variation within and between cells. Furthermore, the separations measured are $>1.2R_0$. Given these two considerations, it is unsurprising that error bars from individual data points are large. This is more acute when the FRET efficiency is very small at low acceptor concentrations, as reflected by the data. This is why robust error analysis was carried out to ascertain significance.

Our principal aim is to determine whether two separations (DI-membrane and DIII-membrane), which were measured in each condition under identical experimental settings, were different or the same. This is our reporter of conformation, and the differences, as acknowledged by the reviewer, are significant.

Other concerns regarded interpretation of results and wording in the text, which are satisfactorily addressed in the revised manuscript.

Reviewer #2 (Remarks to the Author): The authors have answered my questions. I have no further comments.

Reviewer #3 (Remarks to the Author): While the authors responded that the question "how does EGFR receptor density affect the multimerization and phosphorylation?" raised by rev 3, is important, the revised text does little to address this concern. The fact that between two different cell lines known to have different levels of EGFR expression does not immediately lead this reviewer to the conclusion that dimerization is 'independent' of EGFR concentration. Given that so many of the methods used here depend critically on this question, it would have been reassuring if a study using shRNA, inducible or disabled promoters were used to firmly demonstrate this case.

We do not believe the further study suggested is warranted because our results are already consistent between two cell lines that fundamentally differ on their biology: One is a hamster ovary cell line (CHO) and the other a human neoplastic mammary gland cell line (T47D). CHO cells express EGFR alone whilst T47D also express HER2 and HER3. The CHO cells express 10^5 receptors per cell and T47D $<10^4$. One therefore must conclude that the disparity of the CHO and T47D cell models is much larger than one would obtain by varying the number of receptors within the same cell line.

While the DiD-FRET measurements seem to be slightly improved, they still span a very narrow range in acceptor density ...

The acceptor concentration is not narrow but rather typical. We use 0-0.4 acceptors per R_0^2 (for our R_0 of 5.4 nm this equates to 0.014 acceptors per square nm or 14,000 acceptors per square

micrometre). An upper limit of 0.4-0.5 acceptors per R_0^2 is the standard for distinguishing the FRET response to increasing acceptor density of two different datasets without overloading the membrane with probe (See for example Chigaev et al. *Mol. Biol. Cell*, 26 (2014); Wiethoff et al., *J. Biol. Chem.* 277, 44980-44982 (2002); Tynan et al *Mol Cell Biol.* 2011; Needham et al, *Nat Comms* 2016).

Of the 10 conditions analysed in this work, 9 sample throughout the range of 0-0.4 acceptors per R_0^2 . Only in 1 of 9 conditions (Supplementary Fig. 10b, left column, red) the DI-membrane separations sample the range 0-0.2 acceptors per R_0^2 (this is due to experimental variation in the amount of dye absorbed by the membrane). This DI-membrane data-set is however the most abundantly distinguishable from its DIII-membrane counterpart (see Supplementary Fig. 10b, right column, and Supplementary table 1). We also note that a recent DOCA measurement of EGFR by the Jovin group (Ziomkiewicz et al., *Cytometry*, 83A, 794-805 (2013)) successfully used measurements taken exclusively from cells within a narrow range of acceptor density, 0.003-0.004 acceptors per square nm with a 5.4 nm Forster radius, which is considerably below our upper limit.

... the statistical model does not seem valid.

The model we used is uncontroversial. It is the experimentally known ground truth: acceptors in the plane of the membrane and donors in a plane above (see Fig. 4e in submitted manuscript and panel A below from Tynan et al *Mol Cell Biol* 2011). The separations versus acceptor concentration expected from this model were calculated using well understood MonteCarlo techniques (Tynan et al 2011; Needham et al *Nat Comms* 2016).

We would like to point out that in Tynan et al 2011 we considered a more complex model where we modelled close-by donors (dimers) (see panel B above). We used the simplest model here because both models are indistinguishable for separations $>1.2R_0$ (all the distances detected are $>1.2R_0$). Using the more complex model is therefore not warranted.

Shotgun blast data seems to be giving distance measurements with +/- 0.7 nm accuracy?

The reviewer criticises the appearance of the data without any acknowledgement of the statistical significance (as acknowledged by Reviewer 1). Like any equivalent FRET data (see all references

above), our FRET data reflects the intrinsic (and rather typical) variation within and between cells. Furthermore, the separations measured are $>1.2R_0$; thus unsurprisingly error bars from individual data points are large. This is more acute when the FRET efficiency is very small at low acceptor concentrations, as reflected by the data. This is why robust error analysis is required.

The reviewer also mentions accuracy. This is incorrect. The error quoted (e.g. ± 0.7 nm) is the precision of the measurement not its accuracy. We used statistical analysis (bootstrap regression) to objectively assess this precision. The quoted errors arise from comparing the data to an uncontroversial model generated by simulating a donor above a plane of acceptors of increasing density.

Each bootstrap distribution was generated by random sampling of the original data with replacement before fitting to the model as described in Tynan et al (2011) and repeating this process 3,000 times (see Supplementary Fig 10, right hand side column). The error we quote is the standard deviation from the bootstrap analysis. This objectively takes into consideration the errors of individual data points. The key consideration that the reviewer may have missed is that there is a lot of redundancy in the data because all the measurements in each condition measure the same separation.

As stated in our response to Reviewer 1, our aim is to determine whether two separations (DI-membrane and DIII-membrane), which were measured in each condition under identical experimental settings, were different or the same. This is our reporter of conformation. The analysis compares the differences between DI-membrane and DIII-membrane separations with the null hypothesis (both DI-membrane and DIII-membrane separations are the same). It does this 3,000 times (removing data points at random) returning the statistics quoted in Supplementary Table 4.